# Limits to Depth-Efficiencies of Self-Attention

**Yoav Levine, Noam Wies, Or Sharir, Hofit Bata, and Amnon Shashua**
The Hebrew University of Jerusalem

## Abstract

Self-attention architectures, which are rapidly pushing the frontier in natural language processing, demonstrate a surprising depth-**in**efficient behavior: previous works indicate that increasing the internal representation (network width) is just as useful as increasing the number of self-attention layers (network depth). We theoretically predict a width-dependent transition between depth-efficiency and depth-**in**efficiency in self-attention. We conduct systematic empirical ablations on networks of depths 6 to 48 that clearly reveal the theoretically predicted behaviors, and provide explicit quantitative suggestions regarding the optimal depth-to-width allocation for a given self-attention network size. The race towards beyond 1-Trillion parameter language models renders informed guidelines for increasing self-attention depth and width in tandem an essential ingredient. Our guidelines elucidate the depth-to-width trade-off in self-attention networks of sizes up to the scale of GPT3 (which is too deep for its size), and beyond, marking an unprecedented width of 30K as optimal for a 1-Trillion parameter self-attention network.

## 1 Introduction

The golden age of deep learning has popularized the depth-efficiency notion: From an expressiveness standpoint, increasing a neural network's size by adding more layers (deepening) is advantageous relatively to other parameter increase alternatives, such as increasing the dimension of the internal representation (widening). Beyond overwhelming empirical signals for this notion [Simonyan and Zisserman, 2014, He et al., 2016], depth-efficiency was theoretically supported from a variety of angles [Cohen et al., 2016, Eldan and Shamir, 2016, Raghu et al., 2017, Daniely, 2017].

Diminishing returns in the case of very deep networks were mainly attributed to optimization issues, and indeed the alleviation of these issues has allowed network depths to mount from 10s to 100s and beyond [He et al., 2016], enabling deep convolutional networks (ConvNets) to advance the state-of-the-art in computer vision applications. However, as the field matured, a more nuanced perspective emerged. Empirical [Zagoruyko and Komodakis, 2016, Wu et al., 2019] and theoretical [Lu et al., 2017] studies suggest that the interplay between depth and width may be more subtle. Recently, a method for increasing width and depth in tandem ("EfficientNet" by Tan and Le [2019]) has lead to the state-of-the-art on ImageNet while using a ConvNet with a fraction of the parameters used by previous leaders. Our work provides principled guidelines for increasing width and depth in tandem in self-attention networks.

Since the introduction of the Transformer [Vaswani et al., 2017], along with its encoder-only variant, BERT [Devlin et al., 2019], self-attention based deep learning architectures have taken over the field of natural language processing [Liu et al., 2019, Radford et al., 2019, Yang et al., 2019, Raffel et al., 2019a, Clark et al., 2020]. However, in contrast to the depth "arms race" that took place in the ConvNet case, the leading self-attention networks are not much deeper than the original BERT model. In fact, even the strongest self-attention models trained to date, which increased the 0.3B parameter count of BERT-large by factors of 100s to 11B [Raffel et al., 2019a] and 175B [Brown et al., 2020], have only increased its depth by factors of 2 and 4, respectively. The remaining size increase stems from an increase in layer widths, clearly countering the depth-efficiency notion.

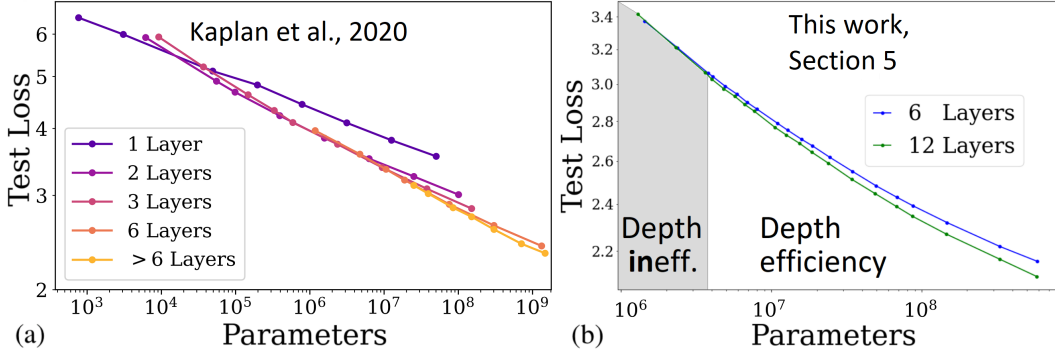

Figure 1: **(a)** An ablation taken from figure 6 in Kaplan et al. [2020], examining the perplexity scores of self-attention networks of varying depths and widths. Experiments on the $L > 6$ curve (yellow) all approximately obey the same improvement trend which depends only on the number of network parameters and not on the depth-to-width ratio. For $L \leq 6$, depth-efficiency is clearly demonstrated, but due to the $L > 6$ curve the authors conclude "depth **in**efficiency" of self attention. **(b)** A representative of our experimental plots, which shows that a transition between depth-efficiency and inefficiency takes place, and that both regimes affect the behavior also at $L > 6$. Figure 2 shows that this trend continues at least up to depth $48$, and figure 3 shows that the transition between regimes grows exponentially with depth, as predicted by our theory.

A recent empirical ablation study by Kaplan et al. [2020] provides support for the above signal. Figure 1(a), taken from this study, leads the authors to conclude that the overall (non-embedding) network size, given by $12 \cdot L \cdot d_x^2$ where $L$ is the number of self-attention layers (network depth) and $d_x$ is the hidden representation dimension (network width), is the main predictor of performance regardless of the depth-to-width ratio. This suggests that depth may not play as crucial a role in self-attention networks as it does in convolutional networks.

In this paper, we theoretically address the above question of the depth-to-width interplay in self-attention network expressivity, and reveal fundamental subtleties in the above picture. **Rather than reinforcing the seemingly plausible hypothesis for the trend in figure 1(a), by which widening a self-attention network is as effective as deepening it, we confirm the contrary.** We show that the operation of stacking self-attention layers is so effective that it quickly saturates a capacity of the network's width. We establish in section 4 the existence of a depth threshold which depends logarithmically on the width $d_x$, denoted $L_{\text{th}}(d_x) \sim \log(d_x)$. Below the threshold, we prove that depth-efficiency takes place in self-attention networks: a network of depth $L \leq L_{\text{th}}(d_x)$ cannot be replicated by a shallower network, unless the latter's width grows double-exponentially with $L$. However, we show that this overwhelming advantage of depth is quickly replaced by a balanced growth. We prove that for self-attention networks with $L > L_{\text{th}}(d_x)$ the ability to model input dependencies increases similarly with depth and width.

After presenting our theoretical analysis in sections 2-4, we provide a thorough empirical evaluation in section 5, which validates our predicted trends for self-attention networks of depths 6 to $48$. Importantly, our theoretical and empirical results provide quantitative guidelines for optimal depth-to-width parameter allocation given a fixed parameter budget (see for example table 1). It seems that popular self-attention architectures at all sizes trained up to GPT3's crossing of the 100B parameter threshold, could generally benefit from deepening, with the appropriate widening (indicated by our guidelines). With that, **our results clearly indicate the importance of widening self-attention networks when aiming for the 1 Trillion parameter mark**. We project the optimal architecture at that size to have depth 95 and width 30K, wider than any self-attention network trained to date.

## 2 The self-attention mechanism

Differentiable attention models in which the output attends over all LSTM-based input representations have been introduced in the context of machine translation [Bahdanau et al., 2014]. Self-attention (also referred to as intra-attention), which relates different inputs to each other, was first employed for machine reading [Cheng et al., 2016], and soon thereafter shown to be useful for a variety of language applications when operating over LSTM-based representations [Parikh et al., 2016, Paulus

| *Borrowed from Brown et al. [2020]* | | **Trained in practice** | | **Optimal by our fit** | |
|---|---|---|---|---|---|
| Model Name | Size in params | Depth ($L$) | Width ($d_x$) | Depth ($L$) | Width ($d_x$) |
| GPT-3 Small | 125M | 12 | 768 | 23 | 555 |
| GPT-3 Medium | 350M | 24 | 1024 | 32 | 886 |
| GPT-3 Large | 760M | 24 | 1536 | 38 | 1220 |
| GPT-3 XL | 1.3B | 24 | 2048 | 42 | 1550 |
| GPT-3 2.7B | 2.7B | 32 | 2560 | 47 | 2110 |
| GPT-3 6.7B | 6.7B | 32 | 4096 | 54 | 3150 |
| GPT-3 13B | 13.0B | 40 | 5140 | 60 | 4200 |
| GPT-3 175B or "GPT-3" | 175.0B | 96 | 12288 | 80 | 13500 |
| Optimal 1-Trillion arch | 1T | – | – | 95 | 30100 |

Table 1: Our projections regarding optimal depth-to-width parameter distribution at self-attention sizes corresponding to huge language models trained in Brown et al. [2020], according to the fit in section 5 (see figure 4 for the statistical uncertainty in these predictions).

et al., 2017, Lin et al., 2017]. Vaswani et al. [2017] were the first to demonstrate that a model based solely on attention, the Transformer, can be better than LSTM based networks. The Transformer's encoder, BERT [Devlin et al., 2019], based entirely on self-attention, has demonstrated unprecedented performance across natural language understanding tasks.

## 2.1 The Transformer encoder architecture

We begin by describing the self-attention operation of the original Transformer, and then in the next subsection we present the modifications made in our analyzed model. Each layer $l \in [L] := \{1, ..., L\}$ of a depth-$L$ Transformer encoder is comprised of two sub-layers. The $H$-headed self-attention sublayer of layer $l$ computes the following function at position $i \in [N]$, over its $N$ inputs $\{\mathbf{x}^{l,j} \in \mathbb{R}^{d_x}\}_{j=1}^{N}$:

$$\mathbf{f}_{\text{SA}}^{l,i}\left(\mathbf{x}^{l,1}, ..., \mathbf{x}^{l,N}\right) = \sum_{j=1}^{N} \sum_{h=1}^{H} SM_j \left\{ {1}/{\sqrt{d_a}} \left\langle W^{\text{Q},l,h}\mathbf{x}^{l,i}, W^{\text{K},l,h}\mathbf{x}^{l,j} \right\rangle \right\} W^{\text{O},l,h} W^{\text{V},l,h}\mathbf{x}^{l,j} \quad (1)$$

where $SM_j \{f(j)\} := e^{f(j)} / \sum_{j'} e^{f(j')}$ is the softmax operation and $\forall h \in [H]$ the learned weights matrices $W^{\text{K},l,h}, W^{\text{Q},l,h}, W^{\text{V},l,h} \in \mathbb{R}^{d_a \times d_x}$ convert the representation from its dimension $d_x$ into the attention dimension $d_a = {d_x}/{H}$, creating Key, Query, and Value representations, respectively. The learned weights matrix $W^{\text{O},l,h} \in \mathbb{R}^{d_x \times d_a}$ converts the attention result back into the representation dimension. The multi-headed self-attention sublayer output in eq. (1), followed by a residual connection and layer-norm [Ba et al., 2016], is inserted into a position-wise feed-forward + ReLU sublayer, such that each layer's output at position $i \in [N]$ is:

$$\mathbf{f}_{\text{Layer}}^{l,i}\left(\mathbf{x}^{l,1}, ..., \mathbf{x}^{l,N}\right) = W^{\text{FF},2} ReLU \left( W^{\text{FF},1} LayerNorm \left( \mathbf{f}_{\text{SA}}^{l,i} + \mathbf{x}^{l,i} \right) \right), \quad (2)$$

where the feed-forward matrices are usually taken to be $W^{\text{FF},1} \in \mathbb{R}^{4d_x \times d_x}, W^{\text{FF},2} \in \mathbb{R}^{d_x \times 4d_x}$, such that the parameter count for an entire layer is $12 \cdot d_x^2$. Finally, the depth-$L$ multi-headed self-attention operation of the Transformer encoder is obtained by a composition of $L$ such layers, *i.e.*, when setting $\forall l \in \{2, ..., L\}, j \in [N] : \mathbf{x}^{l,j} = LayerNorm \left( \mathbf{f}_{\text{Layer}}^{l-1,j} \right)$, with $\mathbf{x}^{1,j}$ denoting the input to the deep self-attention network at position $j$.[1]

## 2.2 The analyzed architecture

We analyze a deep multi-headed self-attention network variant which excludes the layer-norm operation, the softmax normalization, and the ReLU activation (see a thorough discussion on the effect of these relaxations in the next subsection). For cleanliness of presentation, we defer the analysis of the residual connection to the appendix (it bears insignificant impact on our bounds). Specifically, in the analyzed network, each layer $l \in [L]$ computes the following function at position $i \in [N]$ over its inputs $\{\mathbf{x}^{l,j} \in \mathbb{R}^{d_x}\}_{j=1}^{N}$:

$$\mathbf{y}^{l,i}\left(\mathbf{x}^{l,1},...,\mathbf{x}^{l,N}\right) = \sum_{j=1}^{N}\sum_{h=1}^{H}\left\langle W^{\mathrm{Q},l,h}\mathbf{x}^{l,i}, W^{\mathrm{K},l,h}\mathbf{x}^{l,j}\right\rangle W^{\mathrm{O},l,h}W^{\mathrm{V},l,h}\mathbf{x}^{l,j}, \tag{3}$$

where the Feed-Forward matrices can be now effectively embedded within $W^{\mathrm{O},l,h}$. Our analysis below treats a deep multi-headed self-attention network that is attained by a concatenation of $L$ such layers. Importantly, the resultant "linearized" network form, where activations and normalizations are removed, is by no means a linear mapping over the network input – every layer integrates 3 copies of its input in the above non-linear fashion. By recursively applying eq. (3) $L$ times we attain the analyzed depth-$L$ self-attention network.

We denote the function realized by a network with embedding dimension $d_x$ and $H$ attention heads per layer at output location $i \in [N]$ by:

$$\mathbf{y}^{i,L,d_x,H,\Theta}\left(\mathbf{x}^1,...,\mathbf{x}^N\right) := \sum_{j_1,...,j_C=1}^{N}\mathbf{g}^L\left(\mathbf{x}^i,\mathbf{x}^{j_1},...,\mathbf{x}^{j_C}\right), \tag{4}$$

where $\Theta$ denotes all $4LH$ learned weight matrices: $\forall(l,h) \in [L] \otimes [H] : W^{\mathrm{K},l,h}, W^{\mathrm{Q},l,h}, W^{\mathrm{V},l,h} \in \mathbb{R}^{d_a \times d_x}$, and $W^{\mathrm{O},l,h} \in \mathbb{R}^{d_x \times d_a}$, and the function $\mathbf{g}^L$ is a placeholder, fully detailed in the appendix, which integrates $C = \frac{3^L - 1}{2}$ different input vectors. In the following subsection, we comment on the differences between the Transformer encoder architecture described in eqs. (1) and (2) and the self-attention architecture presented in eqs. (3) and (4).

### 2.3 Relaxations

Empirical evidence indicates that while the ReLU activations and softmax normalization contribute to performance, the basic mechanism in eqs. (3) and (4) above captures the defining self-attention characteristic of integrating the inputs with each other in a flexible manner:

*The ReLU activation relaxation*: Press et al. [2019] demonstrate that a "self-attention first" BERT variant that first performs all of the self-attention operations (eq. (1)) consecutively, and only then performs all of the position-wise feed-forward+ReLU operations, achieves comparable language modeling performance relatively to the Baseline, which takes the regular approach of interleaving these functionalities (*i.e.*, concatenating the BERT's layer described in eq. (2)). They report that the interleaved Baseline achieves a perplexity score of $18.63 \pm 0.26$ on the WikiText-103 test [Merity et al., 2016] when averaged over 5 random seeds, while the "self-attention first" model achieves a perplexity score of $18.82$ on this test set. The best pre-Transformer perplexity result on the WikiText-103 test, reported by an LSTM-based architecture, was $29.2$ [Rae et al., 2018]. Since ReLU and feed-forward do not mix different locations, this outcome directly implies that the self-attention mechanism itself provides all of the elaborate input integration which differentiates BERT from previous architectures.

*The softmax normalization relaxation*: Initially, an intuitive interpretation of attention as distributing "fractions" of an overall attention budget among inputs was given to its actual operation of dynamically linking input and output locations. The intuitive interpretation, tightly linked to the need to transform the Key/Query similarity score into a distribution, has been recently challenged, as a growing body of work shows that the attention weights distribution does not directly correlate with predictions [Jain and Wallace, 2019, Pruthi et al., 2019, Brunner et al., 2020]. Moreover, Richter and Wattenhofer [2020] recently point out undesirable traits of the softmax operation, demonstrating that its property of confining the outcome to the convex hull of its inputs unnecessarily limits the expressibility of the self-attention mechanism. They experiment on a suite of synthetic tasks with a BERT variant in which the softmax normalization is removed, and find it to perform on par on almost all examined tasks. When replacing the softmax with other normalizations they report improvements. Finally, completely linearized attention (softmax removed) was employed on real tasks as means of reducing costs, since the softmax operation cost scales with the input size [de Brébisson and Vincent, 2016, Wang et al., 2020].

The goal of the above points is not to advocate modifications in BERT's non-linearity or normalization operations (we leave that to other works), but to note that while these are under examination and are susceptible for alteration, the connectivity of self-attention, manifested by eqs. (3) and (4), is the core mechanism driving its functionality. Our results, to be presented in section 4, demonstrate how

conclusions drawn by directly analyzing this mechanism accord with the operation of commonly employed self-attention networks.

# 3 A measure of capacity for modeling input dependencies

In this section, we introduce the separation rank of the function realized by a self-attention network as a measure that quantifies its ability to model dependencies between subsets of its variable set $\{\mathbf{x}^j\}_{j=1}^N$. We will use this measure in order to establish the two depth-efficiency/ inefficiency regimes in self-attention. The separation rank, introduced in Beylkin and Mohlenkamp [2002] for high-dimensional numerical analysis, was employed for various applications, *e.g.*, chemistry [Harrison et al., 2003], particle engineering [Hackbusch, 2006], and machine learning [Beylkin et al., 2009]. Importantly, the separation rank has been established as a measure of dependencies modeled by deep convolutional and recurrent networks w.r.t. their inputs [Cohen and Shashua, 2017, Levine et al., 2018a,b].

Let $(A, B)$ be a partition of the input locations, *i.e.*, $A$ and $B$ are disjoint subsets of $[N]$ whose union gives $[N]$. The separation rank of a function $y(\mathbf{x}^1, \ldots, \mathbf{x}^N)$ w.r.t. partition $(A, B)$, is the minimal number of summands that together sum up to equal $y$, where each summand is *multiplicatively separable w.r.t.* $(A, B)$, *i.e.*, is equal to a product of two functions – one that intakes only inputs from one subset $\{\mathbf{x}^j : j \in A\}$, and another that intakes only inputs from the other subset $\{\mathbf{x}^j : j \in B\}$. Formally, the *separation rank* of $y : (\mathbb{R}^{d_x})^N \to \mathbb{R}$ w.r.t. the partition $(A, B)$ is defined as follows:

$$sep(y; A, B) := \min \left\{ R \in \mathbb{N} \cup \{0\} : \exists g_1 \ldots g_R : (\mathbb{R}^{d_x})^{|A|} \to \mathbb{R}, g_1' \ldots g_R' : (\mathbb{R}^{d_x})^{|B|} \to \mathbb{R} \ \ s.t. \right. \tag{5}$$

$$\left. y\left(\mathbf{x}^1, \ldots, \mathbf{x}^N\right) = \sum_{r=1}^R g_r \left(\{\mathbf{x}^j : j \in A\}\right) g_r' \left(\{\mathbf{x}^j : j \in B\}\right) \right\}$$

If the separation rank of a function w.r.t. a partition of its input is equal to $1$, the function is separable, meaning it cannot take into account consistency between the values of $\{\mathbf{x}^j\}_{j \in A}$ and those of $\{\mathbf{x}^j\}_{j \in B}$. In a statistical setting, if $y$ is a probability density function, this would mean that $\{\mathbf{x}^j\}_{j \in A}$ and $\{\mathbf{x}^j\}_{j \in B}$ are statistically independent. The higher $sep(y; A, B)$ is, the farther $y$ is from this situation, *i.e.* the more it models dependency between $\{\mathbf{x}^j\}_{j \in A}$ and $\{\mathbf{x}^j\}_{j \in B}$, or equivalently, the stronger the correlation it induces between the inputs indexed by $A$ and those indexed by $B$.

The fixed connectivity of ConvNets has been shown to yield high separation ranks w.r.t. partitions which separate neighboring inputs (*e.g.*, where all odd positions are in $A$ and all even positions are in $B$), while suffering from low separation ranks w.r.t. partitions which separate distant inputs (*e.g.*, where $A = 1, ..., N/2$ and $B = N/2 + 1, ..., N$). We establish a qualitatively different trait for self-attention networks, which treat all balanced partitions alike, *i.e.*, $\forall A \cup B = [N], \tilde{A} \cup \tilde{B} = [N], \ \ s.t. \ |A|, |B|, |\tilde{A}|, |\tilde{B}| = N/2: sep(y_p^{i,L,d_x,H,\Theta}; A, B) = sep(y_p^{i,L,d_x,H,\Theta}; \tilde{A}, \tilde{B})$ (see proof in the appendix). Accordingly, we will omit the specification of the partition in future uses, denoting $sep(y_p^{i,L,d_x,H,\Theta})$ as the separation rank of $y_p^{i,L,d_x,H,\Theta}$ w.r.t. any balanced partition of the inputs.

This result accords with the intuition regarding the flexibility of the attention mechanism – it does not integrate the input in a predefined pattern like convolutional networks, but dynamically learns to correlate any inter-dependent subsets of the inputs. Natural text exhibits non-smooth non-local dependency structures, as correlations between input segments can abruptly rise and decay with distance. The fact that self-attention facilitates all correlation patterns equally poses it as a more natural architecture for language modeling related tasks. Convolutional networks, with their local connectivity, may have the right inductive bias for imagery data, but partitions unfavored by them may reflect more erratic correlations that are nonetheless relevant for natural language inputs.

However, the above property of indifference to the input partition is not enough for succeeding at tasks with elaborate input dependencies, since a function with equally low separation ranks for all input partitions has limited ability to model such dependencies. In the following section, we analyze how different architectural parameters affect the ability of self-attention networks to correlate their inputs, and by bounding their separation ranks, we establish the different depth-efficiency regimes in self-attention networks.

# 4 The effect of depth in self-attention networks

In this section, we present tight bounds on the separation rank of self-attention networks, which reveal two qualitatively different regimes. In the first regime of $L < \log_3(d_x)$, analyzed in subsection 4.1, we establish that deepening is clearly preferable to widening. In the second regime of $L > \log_3(d_x)$,

analyzed in subsection 4.2, we show that deepening and widening play a similar role in enhancing the expressiveness self-attention networks.

## 4.1 Depth-efficiency in self-attention

The recursive structure of deep self-attention hints at an exponential increase of input mixing with depth: The output of each layer is introduced 3 times into the Key/Query/Value computation made by the subsequent layer. In this subsection, we formalize this intuition for self-attention networks of sufficient width, $d_x > 3^L$. Theorem 1 below bounds the separation rank of such networks. Subsequent to its statement and brief outline of its proof, we explicitly show in corollary 1 the implied double-exponential requirement from a bounded depth network attempting to replicate a deeper one.

**Theorem 1.** *For $p \in [d_x]$, let $y_p^{i,L,d_x,H,\Theta}$ be the scalar function computing the pth entry of an output vector at position $i \in [N]$ of the depth-$L$ self-attention network with embedding dimension $d_x$ and $H$ attention heads per layer, defined in eqs. (3) and (4). Let $sep(y_p^{i,L,d_x,H,\Theta})$ be its separation rank (section 3). If $L, d_x$ obey $L < \log_3(d_x)$, then the following holds almost everywhere in the network's learned parameter space,* i.e. *for all values of the weight matrices (represented by $\Theta$) but a set of Lebesgue measure zero:*

$$3^{L-2}\left(\log_3(d_x - H) + a\right) \leq \log_3\left(sep(y_p^{i,L,d_x,H,\Theta})\right) \leq \frac{3^L - 1}{2}\log_3(d_x + H) \qquad (6)$$

*with $a = -L + [2 - \log_3 2]$. (note that $\log_3(d_x - H) + a > 0$ in this regime of $L < \log_3(d_x)$).*

Theorem 1 bounds the separation rank of a deep self-attention network of sufficient width between two functions that grow double-exponentially with depth and polynomially with width, tightly describing its behavior w.r.t. depth and width. Because equivalence cannot hold between two functions of different separation ranks, the above result implies a double-exponential requirement from the width of a shallow network attempting to replicate the deep one, and clear depth-efficiency holds:

**Corollary 1.** *With probability $1$, the function realized upon randomization of the weights of a deep self-attention network defined in eqs. (3) and (4) with depth $L^{deep}$ and width $d_x^{deep} > 3^{L^{deep}}$, may only be realized by a shallower network with depth $L^{shallow} = {}^{L^{deep}}/d$ and width $d_x^{shallow} = w d_x^{shallow}$, where $d > 1, w > 1$ (i.e., the deep network is deeper by a factor of $d$ and the shallow network is wider by a factor of $w$), if the following holds:*
$$w \propto \exp(\exp(d)).$$

## 4.2 Depth in-efficiency in self-attention

Beyond establishing depth-efficiency in early self-attention layers, the above analysis sheds light on the contribution of a self-attention network's depth to its ability to correlate input subsets. The separation rank (w.r.t. any partition) of a single layer, given by eq. (3), is only linear in $H$ and $d_x$, showcasing a limitation of the class of functions realized by single self-attention layers to model elaborate input dependencies. Theorem 1 quantifies the double exponential growth of this capacity measure with the number of stacked self-attention layers. The following theorem shows that this growth is capped by the dimension of the internal representation:

**Theorem 2.** *For $y_p^{i,L,d_x,H,\Theta}$ as defined in theorem 1, if $L > \log_3(d_x)$, then the following holds almost everywhere in the network's learned parameter space,* i.e. *for all values of the weight matrices (represented by $\Theta$) but a set of Lebesgue measure zero:*

$$\frac{1}{2}d_x \cdot L + b_1 + b_2 \leq \log_3\left(sep(y_p^{i,L,d_x,H,\Theta})\right) \leq 2d_x \cdot L + c_1 + c_2 \qquad (7)$$

*with corrections on the order of $L$: $b_1 = -L\left(\frac{H}{2} + 1\right)$, $c_1 = L$, and on the order of $d_x \log_3(d_x)$: $b_2 = -d_x\left(1 + \frac{1}{2}\log_3\left(\frac{d_x - H}{2}\right)\right)$, $c_2 = -2d_x \cdot \log_3 {}^{d_x}/2\sqrt{2e} + \log_3 d_x$.*

Theorem 2 states that when the network's depth passes a width dependent threshold, the separation rank turns from increasing polynomially with width and double-exponentially with depth to increasing-exponentially with width and depth together. Thus, while an increase in network size increases its capacity to model input dependencies, our result shows that there is no longer a clear cut advantage of depth in this respect:

**Corollary 2.** *Let $\mathbf{y}^{deep}$ denote the function realized by a deep self-attention network at any output location $i \in [N]$, defined in eqs. (3) and (4) with depth and width denoted $L^{deep}, d_x^{deep}$ such that*

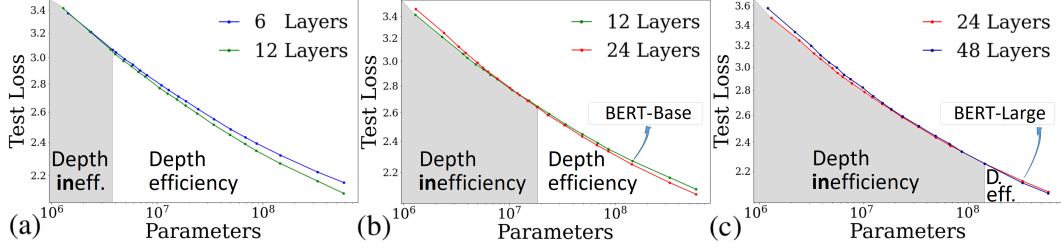

Figure 2: An experimental validation of the existence of the two depth-efficiency/**in**efficiency regimes for common self-attention networks. The transition between regimes, occurs in exponentially larger network sizes as the shallower network gets deeper, in agreement with our theory (see figure 3).

$L^{deep} > \log_3 d_x^{deep}$. Denote $\beta_1 := \frac{\log_3 d_x^{deep}}{L^{deep}} < 1$. Then, there exists $\beta_2 = O(log(H) \cdot log(d_x^{deep}) \cdot log(L^{deep}))$ such that the function realized by a network of depth: $L^{shallow} = \beta_1 \cdot L^{deep} + \beta_2$, and width: $d_x^{shallow} = 3^{\beta_2} d_x^{deep}$, denoted $\mathbf{y}^{shallow}$, has higher separation rank, i.e.:

$$sep(y_p^{shallow}) > sep(y_{p'}^{deep}) \quad ; \quad where \; p, p' \in [d_x] \tag{8}$$

Corollary 2, which follows from theorems 1 and 2, shows that the separation rank of a function realized by a self-attention network of arbitrary depth $L > \log_3(d_x)$ can be surpassed by a shallower network of polynomial width, contrarily to the established behavior for networks of depth $L < \log_3(d_x)$.

We leave it as an open conjecture that a polynomially sized shallower network can exactly replicate the operation of a deeper network in this regime. With that, we point out that a variety of results which directly bound different complexity measures of deep networks have been put forward, shedding light on their operation [Montufar et al., 2014, Bianchini and Scarselli, 2014, Raghu et al., 2017, Serra et al., 2017, Inoue, 2019]. Bounds on the separation rank have been used to explain the operation of more veteran architectures, and we find them to be particularly relevant in the case of self-attention: this complexity measure quantifies the amount of input inter-dependency induced by the network, directly reflecting a widespread intuition on the success behind the self-attention mechanism.

## 5 Depth-efficiency regimes in common self-attention networks

In the previous sections, we analyzed a simplified version of self-attention networks (described in section 2). For this class, we proved the existence of the two different depth-efficiency/**in**efficiency regimes in self-attention networks, and further quantified the transition point to be exponential in network width (and accordingly in network size). In this section, we demonstrate that our theoretical predictions are manifested in common self-attention networks: the experiments below were conducted over common self-attention architectures which include all operations that were omitted in our theoretical analysis. The training apparatus details are given in the appendix.

### 5.1 Distinct depth-efficiency regimes in self-attention

Figure 2 shows that the predicted devision into two depth-efficiency/**in**efficiency regimes indeed takes place in common self-attention architectures. When comparing depths $(L^{\text{shallow}}, L^{\text{deep}}) = \{(6, 12), (12, 24), (24, 48)\}$, a qualitatively different depth-efficiency behavior is observed as the network size varies. For smaller network sizes, deepening is not favorable over widening. Our theoretical analysis predicts this, showing that when the width of the deeper network is not large enough it can not use its excess layers efficiently. However, when the network's size is increased by widening, a transition into the depth-efficiency regime is clearly demonstrated: for the same parameter budget the deeper network performs better. Once the deeper network becomes wide enough, such that the depth threshold for depth-efficiency surpasses $L^{\text{shallow}}$, it is significantly more expressive.

### 5.2 Transition between regimes depends exponentially on depth

Importantly, beyond a qualitative match to the two predicted depth-efficiency/**in**efficiency behaviors, the experiments corroborate our prediction for an exponential dependence of the "depth-efficiency width" — the width for which a network becomes depth-efficient — on the network's depth. By quantifying this exponential behavior we attain practical guidelines for depth-to-width parameter allocation in a self-attention network of a given size (figure 4).

Per network depth, we examine the width in which it diverges from the subsequent trained depth, *i.e.*, we examine the following pairs of trained adjacent depths: $(6, 12), (12, 18), (18, 24), (24, 30), (30, 36), (36, 48)$. For each pair, we estimate the shallower network's transition width (marking the crossing between gray and white areas in figure 2) as the

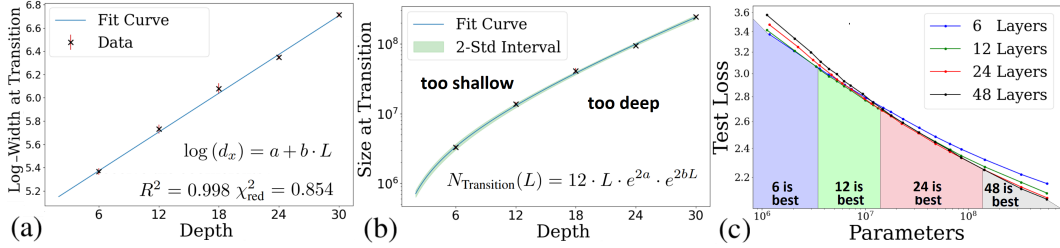

Figure 3: **(a)** A fit of the predicted exponential dependence of network width on its depth at the transition point between the depth-efficiency/inefficiency regimes. **(b)** The network size at the transition between regimes $N_{\text{Transition}}$ as a function of network depth. The green area marks an interval of $2\Delta N_{\text{Transition}}$ as calculated in eq. 9. Architectures to the top-left of the curve can be improved by deepening. **(c)** The color in each range of network sizes corresponds to the color of the depth reaching the minimal loss in this range. This implies that architectures to the bottom-right of the curve in figure (b) can be improved by widening.

average of its width in two points: the first point in which the shallower network under-performs in a statistically significant manner (see standard deviation estimation in the appendix), and the point to its left in which the performance of the two is not distinguishable. We take the empirical error of this estimation to be the distance between the two points.

Our theoretical results in section 4 predict that the above empirically estimated transition should occur when the shallower network's width $d_x$ is exponential in its depth $L$. Accordingly, we fit a linear dependence of the log of the width on the depth and receive the fit coefficients $(a, b)$: $\log(d_x) = a + b \cdot L$. The linear fit, shown in Figure 3(a) yields measures of $R^2 = 0.998$ and $\chi^2_{\text{red}} = 0.854$. These measures imply a good compatibility of the theoretically predicted dependence to the measurements, and further reinforce the practical use we make of the fit parameters $a$ and $b$ hereinafter, for predicting the network size for which the regime transition occurs per depth.

Specifically, we insert $d_x = e^a \cdot e^{bL}$ into the dependence $N = 12 \cdot L \cdot d_x^2$ and calculate the transition size and its propagated uncertainty as:

$$N_{\text{Transition}}(L) = 12 \cdot L \cdot e^{2a} \cdot e^{2bL} \tag{9}$$

$$\Delta N_{\text{Transition}}(L) = \sqrt{\left(\frac{dN}{da}\right)^2 \sigma_a^2 + \left(\frac{dN}{db}\right)^2 \sigma_b^2 + 2\left(\frac{dN}{da}\right)\left(\frac{dN}{db}\right)\sigma_{ab}}$$

with the fit parameters given by:

$$\begin{pmatrix} a & b \end{pmatrix} = \begin{pmatrix} 5.039 \pm 0.030 & 5.55 \cdot 10^{-2} \pm 1.3 \cdot 10^{-3} \end{pmatrix} \tag{10}$$

$$\begin{pmatrix} \sigma_a^2 & \sigma_{ab} \\ \sigma_{ab} & \sigma_b^2 \end{pmatrix} = \begin{pmatrix} 9.4 \cdot 10^{-4} & -3.74 \cdot 10^{-5} \\ -3.74 \cdot 10^{-5} & 1.7 \cdot 10^{-6} \end{pmatrix}$$

Figure 3(b) shows the empirical transition sizes per depth on top of the projection and its error, calculated by eq. 9 with the fit parameters in eq. 10. Networks to the left of the curve are too shallow given their parameter budget, and can be improved by deepening at the expense of their width.

### 5.3 "Width-efficiency" and practical implications

Our experiments reveal an empirical phenomenon that was not predicted by our theory. We established in section 4 that depth does not have an advantage when the width is too small, but our bounds do not separate wider networks from deeper ones in this depth-inefficiency regime. A surprising phenomenon is seen in figures 2(b,c): for small enough network sizes, deeper self-attention networks perform *worse* than shallow ones. We leave a theoretical treatment of this regime for future work.

The above "width-efficiency" empirical phenomenon leads to an important observation: for a given network size, a certain network can be too shallow, as we predicted theoretically and corroborated empirically above, but it can also be **too deep**. In other words, the region to the right of the fit curve in figure 3(b) includes networks that can be improved by widening at the expense of their depth. This implies that rather than representing a minimal depth per given self-attention network size, the curve in figure 3(b) represents the area of an **optimal depth** per network size. We provide a demonstration of this idea in figure 3(c), which clearly shows that when comparing networks of

depths $L = 6, 12, 24, 48$, each one would be best to use in a different range of network sizes (the color in each range corresponds to the best performing depth in that range).

Beyond reflecting our theoretical predictions, the fit in figure 3 can be used to project beyond the scope of our experiments in order to shed light on architecture design decisions made for much larger self-attention networks, like the contemporary huge Transformer-based language models [Brown et al., 2020, Raffel et al., 2019b, Rosset, 2020]. Figure 4 shows the extrapolation of the fitted function and the uncertainty up to networks of depth 100. Notably, despite the uncertainty growing as the scope extends, $\frac{\Delta N_{\text{Transition}}(L=100)}{N_{\text{Transition}}(L=100)} = 0.2$, *i.e.*, the predictions for the optimal network size in the $L = 100$ case are likely to be accurate up to $20\%$ of the predicted size, yielding meaningful and unforeseen practical implications.

For example, when examining the architecture of GPT3, the deepest self-attention network trained to date with 96 layers, we get $N_{\text{Transition}}(96) = 1.17 \pm 0.23 \cdot 10^{12}$, or over a Trillion parameters. This places GPT3 with its 175B parameters significantly below our fit, suggesting that it may be too deep given its parameter budget. In fact, the optimal depth for GPT3's size is predicted to be $L = 80$, since $N_{\text{Transition}}(80) = 1.65 \pm 0.25 \cdot 10^{11}$. Table 1 includes further suggestion for huge models following our fit. With high certainty given our experimental data, the optimal model size increase towards 1 Trillion parameter models and beyond is via widening.

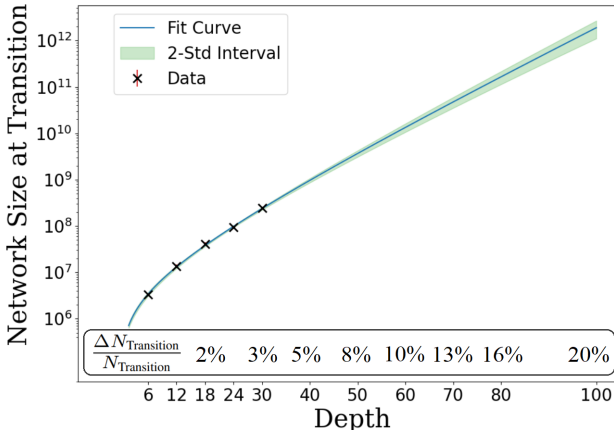

Figure 4: An extrapolation of the function in eq. 9 to sizes that are beyond the scope of our experiments.

## 6 Discussion

An apparent "depth-*in*efficiency" of self-attention networks was pointed out by prior works – in contrast to other successful deep learning architectures, in the case of self-attention there does not seem to be a clear advantage to deepening vs. widening. Our theoretical analysis clearly reflects this behavior in one parameter setting, but suggests an important nuance regarding its origins, while predicting a separate "depth-efficiency" regime in another parameter setting. Rather than an obvious explanation for the observed depth inefficiency, by which the self-attention mechanism does not benefit much from the operation of compounding, our analysis strongly points at the converse: self-attention is so effective at integrating its inputs, that it very quickly reaches saturation in the amount of dependencies that can be supported by the representation dimension.

For early self-attention compounding, we prove a rapid growth in expressiveness with depth, and specifically in the ability to flexibly correlate input locations, which can not be accounted for by reasonable widening. However, our analysis pinpoints a transition in which the capacity of width to support the above rapid growth exhausts. Thus, when the width of a self-attention network is not large enough, deepening and widening become equivalent in terms of expressiveness.

The experiments presented in section 5 reveal a qualitative and quantitative match to our theoretical predictions. Beyond reinforcing the validity of our theoretical interpretation, our comprehensive experimental setup allowed us to extrapolate and project depth-to-width trade-offs in huge self-attention networks, that are currently being trained as powerful language models. For example, GPT3, the deepest self-attention network trained to date with 96 layers, has matched this depth with an unprecedented width of 12K. However, our projections clearly show that for this number of layers the network should be much wider. **In fact, the logarithmic dependence that we establish between the optimal depth and width clearly dictates that size increase should be mainly via widening from this point ( $\sim$ 100B models) onwards.** This is good news from an engineering perspective: width can be increased more efficiently than depth in terms of parallelization. The high price tag on these architectures, along with the pressing race towards 1-Trillion parameter models and beyond, make such informed guidelines an essential ingredient. Indeed, we view our work as part of an effort to provide timely interpretations as feedback for the tremendous empirical pull in our field.

## Broader Impact

Our work aims at providing fundamental guidelines which can assist all fields that employ Transformer-based architectures to use more efficient models. This way, these fields can achieve their goals while consuming less resources.

## Acknowledgments

We thank Daniel Jannai for assistance in the experiments, and Jared Kaplan for the permission to use the figure in Kaplan et al. [2020]. This research was supported by the ERC (European Research Council) and the ISF (Israel Science Foundation). Experiments were performed with Cloud TPUs and supported by Google's TensorFlow Research Cloud (TFRC). Yoav Levine was supported by the Israel Academy of Sciences Adams fellowship.

Jimmy Lei Ba, Jamie Ryan Kiros, and Geoffrey E Hinton. Layer normalization. *arXiv preprint arXiv:1607.06450*, 2016.

Dzmitry Bahdanau, Kyunghyun Cho, and Yoshua Bengio. Neural machine translation by jointly learning to align and translate. *arXiv preprint arXiv:1409.0473*, 2014.

Gregory Beylkin and Martin J Mohlenkamp. Numerical operator calculus in higher dimensions. *Proceedings of the National Academy of Sciences*, 99(16):10246–10251, 2002.

Gregory Beylkin, Jochen Garcke, and Martin J Mohlenkamp. Multivariate regression and machine learning with sums of separable functions. *SIAM Journal on Scientific Computing*, 31(3):1840–1857, 2009.

Monica Bianchini and Franco Scarselli. On the complexity of neural network classifiers: A comparison between shallow and deep architectures. *Neural Networks and Learning Systems, IEEE Transactions on*, 25(8): 1553–1565, 2014.

Tom B Brown, Benjamin Mann, Nick Ryder, Melanie Subbiah, Jared Kaplan, Prafulla Dhariwal, Arvind Neelakantan, Pranav Shyam, Girish Sastry, Amanda Askell, et al. Language models are few-shot learners. *arXiv preprint arXiv:2005.14165*, 2020.

Gino Brunner, Yang Liu, Damian Pascual Ortiz, Oliver Richter, Massimiliano Ciaramita, and Roger Wattenhofer. On identifiability in transformers. 2020.

Jianpeng Cheng, Li Dong, and Mirella Lapata. Long short-term memory-networks for machine reading. *arXiv preprint arXiv:1601.06733*, 2016.

Kevin Clark, Minh-Thang Luong, Quoc V. Le, and Christopher D. Manning. Electra: Pre-training text encoders as discriminators rather than generators. In *International Conference on Learning Representations*, 2020. URL https://openreview.net/forum?id=r1xMH1BtvB.

Nadav Cohen and Amnon Shashua. Inductive bias of deep convolutional networks through pooling geometry. In *5th International Conference on Learning Representations (ICLR)*, 2017.

Nadav Cohen, Or Sharir, and Amnon Shashua. On the expressive power of deep learning: A tensor analysis. *Conference On Learning Theory (COLT)*, 2016.

Amit Daniely. Depth separation for neural networks. *arXiv preprint arXiv:1702.08489*, 2017.

Alexandre de Brébisson and Pascal Vincent. A cheap linear attention mechanism with fast lookups and fixed-size representations. *arXiv preprint arXiv:1609.05866*, 2016.

Jacob Devlin, Ming-Wei Chang, Kenton Lee, and Kristina Toutanova. BERT: pre-training of deep bidirectional transformers for language understanding. In Jill Burstein, Christy Doran, and Thamar Solorio, editors, *Proceedings of the 2019 Conference of the North American Chapter of the Association for Computational Linguistics: Human Language Technologies, NAACL-HLT 2019, Minneapolis, MN, USA, June 2-7, 2019, Volume 1 (Long and Short Papers)*, pages 4171–4186. Association for Computational Linguistics, 2019. doi: 10.18653/v1/n19-1423. URL https://doi.org/10.18653/v1/n19-1423.

Ronen Eldan and Ohad Shamir. The power of depth for feedforward neural networks. In *Conference on learning theory*, pages 907–940, 2016.

Wolfgang Hackbusch. On the efficient evaluation of coalescence integrals in population balance models. *Computing*, 78(2):145–159, 2006.

Robert J Harrison, George I Fann, Takeshi Yanai, and Gregory Beylkin. Multiresolution quantum chemistry in multiwavelet bases. In *Computational Science-ICCS 2003*, pages 103–110. Springer, 2003.

Kaiming He, Xiangyu Zhang, Shaoqing Ren, and Jian Sun. Deep residual learning for image recognition. In *Proceedings of the IEEE Conference on Computer Vision and Pattern Recognition*, pages 770–778, 2016.

K. Inoue. Expressive numbers of two or more hidden layer relu neural networks. In *2019 Seventh International Symposium on Computing and Networking Workshops (CANDARW)*, pages 129–135, 2019.

Sarthak Jain and Byron C. Wallace. Attention is not explanation. In Jill Burstein, Christy Doran, and Thamar Solorio, editors, *Proceedings of the 2019 Conference of the North American Chapter of the Association for Computational Linguistics: Human Language Technologies, NAACL-HLT 2019, Minneapolis, MN, USA, June 2-7, 2019, Volume 1 (Long and Short Papers)*, pages 3543–3556. Association for Computational Linguistics, 2019. doi: 10.18653/v1/n19-1357. URL `https://doi.org/10.18653/v1/n19-1357`.

Jared Kaplan, Sam McCandlish, Tom Henighan, Tom B Brown, Benjamin Chess, Rewon Child, Scott Gray, Alec Radford, Jeffrey Wu, and Dario Amodei. Scaling laws for neural language models. *arXiv preprint arXiv:2001.08361*, 2020.

Yoav Levine, Or Sharir, Alon Ziv, and Amnon Shashua. Benefits of depth for long-term memory of recurrent networks. *(ICLR 2018) International Conference on Learning Representations workshop*, 2018a.

Yoav Levine, David Yakira, Nadav Cohen, and Amnon Shashua. Deep learning and quantum entanglement: Fundamental connections with implications to network design. In *6th International Conference on Learning Representations (ICLR)*, 2018b.

Zhouhan Lin, Minwei Feng, Cicero Nogueira dos Santos, Mo Yu, Bing Xiang, Bowen Zhou, and Yoshua Bengio. A structured self-attentive sentence embedding. *arXiv preprint arXiv:1703.03130*, 2017.

Yinhan Liu, Myle Ott, Naman Goyal, Jingfei Du, Mandar Joshi, Danqi Chen, Omer Levy, Mike Lewis, Luke Zettlemoyer, and Veselin Stoyanov. Roberta: A robustly optimized BERT pretraining approach. *CoRR*, abs/1907.11692, 2019. URL `http://arxiv.org/abs/1907.11692`.

Zhou Lu, Hongming Pu, Feicheng Wang, Zhiqiang Hu, and Liwei Wang. The expressive power of neural networks: A view from the width. In *Advances in neural information processing systems*, pages 6231–6239, 2017.

Stephen Merity, Caiming Xiong, James Bradbury, and Richard Socher. Pointer sentinel mixture models. *arXiv preprint arXiv:1609.07843*, 2016.

Guido F Montufar, Razvan Pascanu, Kyunghyun Cho, and Yoshua Bengio. On the number of linear regions of deep neural networks. In *Advances in neural information processing systems*, pages 2924–2932, 2014.

Ankur P Parikh, Oscar Täckström, Dipanjan Das, and Jakob Uszkoreit. A decomposable attention model for natural language inference. *arXiv preprint arXiv:1606.01933*, 2016.

Romain Paulus, Caiming Xiong, and Richard Socher. A deep reinforced model for abstractive summarization. *arXiv preprint arXiv:1705.04304*, 2017.

Ofir Press, Noah A Smith, and Omer Levy. Improving transformer models by reordering their sublayers. *arXiv preprint arXiv:1911.03864*, 2019.

Danish Pruthi, Mansi Gupta, Bhuwan Dhingra, Graham Neubig, and Zachary C Lipton. Learning to deceive with attention-based explanations. *arXiv preprint arXiv:1909.07913*, 2019.

Alec Radford, Jeff Wu, Rewon Child, David Luan, Dario Amodei, and Ilya Sutskever. Language models are unsupervised multitask learners. 2019.

Jack W Rae, Chris Dyer, Peter Dayan, and Timothy P Lillicrap. Fast parametric learning with activation memorization. *arXiv preprint arXiv:1803.10049*, 2018.

Colin Raffel, Noam Shazeer, Adam Roberts, Katherine Lee, Sharan Narang, Michael Matena, Yanqi Zhou, Wei Li, and Peter J Liu. Exploring the limits of transfer learning with a unified text-to-text transformer. *arXiv preprint arXiv:1910.10683*, 2019a.

Colin Raffel, Noam Shazeer, Adam Roberts, Katherine Lee, Sharan Narang, Michael Matena, Yanqi Zhou, Wei Li, and Peter J Liu. Exploring the limits of transfer learning with a unified text-to-text transformer. *arXiv preprint arXiv:1910.10683*, 2019b.

Maithra Raghu, Ben Poole, Jon Kleinberg, Surya Ganguli, and Jascha Sohl Dickstein. On the expressive power of deep neural networks. In *Proceedings of the 34th International Conference on Machine Learning-Volume 70*, pages 2847–2854. JMLR. org, 2017.

Oliver Richter and Roger Wattenhofer. Normalized attention without probability cage. *arXiv preprint arXiv:2005.09561*, 2020.

Corby Rosset. Turing-NLG: A 17-billion-parameter language model by microsoft. https://www.microsoft.com/en-us/research/blog/turing-nlg-a-17-billion-parameter-language-model-by-microsoft/, 2020. Accessed: 2020-04-12.

Thiago Serra, Christian Tjandraatmadja, and Srikumar Ramalingam. Bounding and counting linear regions of deep neural networks. *arXiv preprint arXiv:1711.02114*, 2017.

Karen Simonyan and Andrew Zisserman. Very deep convolutional networks for large-scale image recognition. *arXiv preprint arXiv:1409.1556*, 2014.

Mingxing Tan and Quoc V Le. Efficientnet: Rethinking model scaling for convolutional neural networks. *arXiv preprint arXiv:1905.11946*, 2019.

Ashish Vaswani, Noam Shazeer, Niki Parmar, Jakob Uszkoreit, Llion Jones, Aidan N. Gomez, Lukasz Kaiser, and Illia Polosukhin. Attention is all you need. In Isabelle Guyon, Ulrike von Luxburg, Samy Bengio, Hanna M. Wallach, Rob Fergus, S. V. N. Vishwanathan, and Roman Garnett, editors, *Advances in Neural Information Processing Systems 30: Annual Conference on Neural Information Processing Systems 2017, 4-9 December 2017, Long Beach, CA, USA*, pages 5998–6008, 2017. URL http://papers.nips.cc/paper/7181-attention-is-all-you-need.

Chengwei Wang, Tengfei Zhou, Chen Chen, Tianlei Hu, and Gang Chen. Off-policy recommendation system without exploration. In *Pacific-Asia Conference on Knowledge Discovery and Data Mining*, pages 16–27. Springer, 2020.

Zifeng Wu, Chunhua Shen, and Anton Van Den Hengel. Wider or deeper: Revisiting the resnet model for visual recognition. *Pattern Recognition*, 90:119–133, 2019.

Zhilin Yang, Zihang Dai, Yiming Yang, Jaime G. Carbonell, Ruslan Salakhutdinov, and Quoc V. Le. Xlnet: Generalized autoregressive pretraining for language understanding. In Hanna M. Wallach, Hugo Larochelle, Alina Beygelzimer, Florence d'Alché-Buc, Emily B. Fox, and Roman Garnett, editors, *Advances in Neural Information Processing Systems 32: Annual Conference on Neural Information Processing Systems 2019, NeurIPS 2019, 8-14 December 2019, Vancouver, BC, Canada*, pages 5754–5764, 2019. URL http://papers.nips.cc/paper/8812-xlnet-generalized-autoregressive-pretraining-for-language-understanding.

Sergey Zagoruyko and Nikos Komodakis. Wide residual networks. *arXiv preprint arXiv:1605.07146*, 2016.

## Footnotes

[1]Focusing on the self-attention operation, we omit a description of the input embedding matrix, as well as of the positional embeddings added at the input, which do not affect our analysis given realistic vocabulary sizes.
