[Supplementary Material]

# Limits to Depth Efficiencies of Self-Attention
# Supplementary Material

**Yoav Levine**
The Hebrew University of Jerusalem
yoavlevine@cs.huji.ac.il

**Noam Wies**
The Hebrew University of Jerusalem
noam.wies@cs.huji.ac.il

**Or Sharir**
The Hebrew University of Jerusalem
or.sharir@cs.huji.ac.il

**Hofit Bata**
The Hebrew University of Jerusalem
hofit.bata@cs.huji.ac.il

**Amnon Shashua**
The Hebrew University of Jerusalem
amnons@cs.huji.ac.il

## Contents

# A  Upper bounds on the separation rank

## A.1  The function realized by a deep multi-headed self-attention network

In this subsection, we prove facts on the general structure of the function realized by the analyzed self-attention architecture that will be of use to us in the upcoming proofs. For a cleaner presentation, we will rewrite eq. 3 of the main text in vectorized notation:

$$Y = \sum_{h=1}^{H} W^{\mathrm{O},h} W^{\mathrm{V},h} X X^{T} \left(W^{\mathrm{K},h}\right)^{T} W^{\mathrm{Q},h} X \tag{1}$$

where $X, Y(X) \in \mathbb{R}^{d_x \times N}$ denote matrices respectively holding $\mathbf{x}^j, \mathbf{y}^j \left(\mathbf{x}^1, ..., \mathbf{x}^N\right)$ in their $j$'th column. Similarly treating eq. 4 of the main text, we will denote by $Y^{L,d_x,H,\Theta}(X) \in \mathbb{R}^{d_x \times N}$ the matrix holding $y^{j,L,d_x,H,\Theta}\left(x^1, ..., x^N\right)$ in its $j$'th column.

We begin by proving a lemma that reveals the structure of $\mathbf{g}^L$ presented in eq. 4 of the main text:

**Lemma 1.** *Defining $C(L) := \frac{3^L - 1}{2}$, any depth $L$ composition of the self-attention layers defined in eq. 3 of the main text can be written as:*

$$Y^{L,d_x,H,\Theta} = \sum_{h \in [H]^{[C(L)]}} B^{(0,h)T} M^{(1,h)} \cdots M^{(C(L),h)} A^{(0,h)} X \tag{2}$$

*where $\forall h \in [H]^{[C]}$ $0 \le c \le C(L) : M^{(c,h)} = A^{(c,h)} X X^T B^{(c,h)T}$ and $A^{(c,h)}, B^{(c,h)} \in \mathbb{R}^{d_a \times d_x}$.*

*Proof.* By Induction on $L$. Base case:

$$Y^{(1)}(X) = \sum_{h=1}^{H} \underbrace{W^{\mathrm{O},h}}_{B^T} \underbrace{W^{\mathrm{V},h} X X^T \left(W^{\mathrm{K},h}\right)^T}_{M} \underbrace{W^{\mathrm{Q},h}}_{A} X$$

$$Y^{(L+1)}(X) = \sum_{h=1}^{H} W^{\mathrm{O},h} W^{\mathrm{V},h} Y^{(L)}(X) Y^{(L)}(X)^T \left(W^{\mathrm{K},h}\right)^T W^{\mathrm{Q},h} Y^{(L)}(X)$$

Now, substituting in the induction hypothesis on the structure of $Y^{(L)}(X)$ yields:

$$= \sum_{h=1}^{H} W^{\mathrm{O},h} W^{\mathrm{V},h} \left( \sum_{h_1 \in [H]^{[C(L)]}} B^{(0,h_1)T} M^{(1,h_1)} \cdots M^{(C(L),h_1)} A^{(0,h_1)} X \right)$$

$$\left( \sum_{h_2 \in [H]^{[C(L)]}} X^T A^{(0,h_2)T} M^{(C(L),h_2)T} \cdots M^{(1,h_2)T} B^{(0,h_2)} \right) \left(W^{\mathrm{K},h}\right)^T W^{\mathrm{Q},h}$$

$$\left( \sum_{h_3 \in [H]^{[C(L)]}} B^{(0,h_3)T} M^{(1,h_3)} \cdots M^{(C(L),h_3)} A^{(0,h_3)} X \right)$$

Finally unifying the summations over $h, h_1 h_2, h_3$ to single sum over $[H]^{[C(L) \cdot 3 + 1 = C(L+1)]}$ gives

$$\sum_{h \in [H]^{[C(L+1)]}} \underbrace{W^{\mathrm{O},h} W^{\mathrm{V},h} B^{(0,h)T}}_{\in \mathbb{R}^{d_x \times d_a}} M^{(1,h)} \cdots M^{(C(L),h)} \underbrace{A^{(0,h)} X X^T A^{(0,h)T}}_{\text{in the desired form of } M} M^{(C(L),h)T} \cdots M^{(2,h)T}$$

$$\tag{3}$$

$$\underbrace{M^{(1,h)T} B^{(0,h)} \left(W^{\mathrm{K},h(0)}\right)^T W^{\mathrm{Q},h} B^{(0,h)T} M^{(1,h)}}_{\text{in the desired form of } M} \cdots M^{(C(L),h)} A^{(0,h)} X$$

$$\tag{4}$$

Note that the number of $M$ units, each with a summation on a different index $j \in [N]$, is $3C(L)+1 = C(L+1)$, implying $C(L) = \frac{3^L-1}{2}$ as needed. $\qquad\square$

**Corollary 1.** *Defining* $C(L) := \frac{3^L-1}{2}$, *any depth $L$ composition of $L$ self-attention layers can be written as:*

$$\mathbf{y}^{i,L,d_x,H,\Theta}\left(\mathbf{x}^1,...,\mathbf{x}^N\right) = \sum_{j_1,...,j_C=1}^{N} \mathbf{g}^L\left(\mathbf{x}^i,\mathbf{x}^{j_1},...,\mathbf{x}^{j_C}\right) \tag{5}$$

*Where*

$$\mathbf{g}^L\left(\mathbf{x}^i,\mathbf{x}^{j_1},...,\mathbf{x}^{j_C}\right) := \sum_{h\in[H]^{[C(L)]}}\sum_{r_1,...,r_{C(L)+1}=1}^{d_a}\left[B^{(0,h)}\right]_{r_1,p}\left(\prod_{c=1}^{C(L)}\left\langle A_{r_c}^{(c,h)},\mathbf{x}^{(j_c)}\right\rangle\left\langle B_{r_{c+1}}^{(c,h)},\mathbf{x}^{(j_c)}\right\rangle\right)\left\langle A_{r_{C(L)+1}}^{(0,h)},\mathbf{x}^{(i)}\right\rangle$$

*Proof.* To get the required form, we will use lemma 1 above and write the matrix multiplication in eq. (2) explicitly.

$$M_{r_1,r_2}^{(c,h)} = \sum_{j=1}^{N}\left[A^{(c,h)}X\right]_{r_1,j}\left[X^T B^{(c,h)T}\right]_{j,r_2} = \sum_{j=1}^{N}\left\langle A_{r_1}^{(c,h)},\mathbf{x}^{(j)}\right\rangle\left\langle B_{r_2}^{(c,h)},\mathbf{x}^{(j)}\right\rangle$$

Therefore

$$y_p^{i,L,d_x,H,\Theta}\left(\mathbf{x}^{(1)},...,\mathbf{x}^{(N)}\right) = \sum_{h\in[H]^{[C(L)]}} B_p^{(0,h)T} M^{(1,h)}\cdots M^{(C(L),h)} A^{(0,h)}\mathbf{x}^{(i)}$$

$$= \sum_{j_1,...,j_{C(L)}=1}^{N}\sum_{h\in[H]^{[C(L)]}}\sum_{r_1,...,r_{C(L)+1}=1}^{d_a}$$

$$\left[B^{(0,h)}\right]_{r_1,p}\left(\prod_{c=1}^{C(L)}\left\langle A_{r_c}^{(c,h)},\mathbf{x}^{(j_c)}\right\rangle\left\langle B_{r_{c+1}}^{(c,h)},\mathbf{x}^{(j_c)}\right\rangle\right)\left\langle A_{r_{C(L)+1}}^{(0,h)},\mathbf{x}^{(i)}\right\rangle$$

$\qquad\square$

In the next two subsections, we will use the above lemma 1 to prove the two competing upper bounds on the separation rank of self-attention networks.

## A.2 Proof of the upper bound in theorem 1

In the following theorem, we show how an upper bound on the separation rank is implied by the form of eq. (2) in the statement of lemma 1.

**Theorem 1.** *Defining* $C(L) := \frac{3^L-1}{2}$, *for any depth $L \geq 1$ input size $N > 1$ partition $P \cup Q = [N]$ and output locations $i \in [N]$, $p \in [d_x]$, the following holds:*

$$sep\left(y_p^{i,L,d_x,H,\Theta}, P, Q\right) \leq (H(d_a+1))^{C(L)}$$

*Proof.* We begin by writing the matrix multiplication in eq. (2) explicitly.

$$M_{r_1,r_2}^{(c,h)} = \sum_{j=1}^{N}\left[A^{(c,h)}X\right]_{r_1,j}\left[X^T B^{(c,h)T}\right]_{j,r_2} = \sum_{j\in P}\left\langle A_{r_1}^{(c,h)},\mathbf{x}^{(j)}\right\rangle\left\langle B_{r_2}^{(c,h)},\mathbf{x}^{(j)}\right\rangle+\sum_{j\in Q}\left\langle A_{r_1}^{(c,h)},\mathbf{x}^{(j)}\right\rangle\left\langle B_{r_2}^{(c,h)},\mathbf{x}^{(j)}\right\rangle$$

Therefore, rewriting the summation to be over $\{P_c \in \{P, Q\}\}_{c=1}^{C(L)}$ that correspond to the two partition segments $P/Q$.

$$y_p^{i,L,d_x,H,\Theta}\left(\mathbf{x}^{(1)}, ..., \mathbf{x}^{(N)}\right) = \sum_{h \in [H]^{[C(L)]}} B_p^{(0,h)T} M^{(1,h)} \cdots M^{(C(L),h)} A^{(0,h)} \mathbf{x}^{(i)}$$

$$= \sum_{h \in [H]^{[C(L)]}} \sum_{r_1, ..., r_{C(L)+1}=1}^{d_a} \sum_{P_1, ..., P_{C(L)} \in \{P, Q\}}$$

$$B_{r_1,p}^{(0,h)} \left( \prod_{c=1}^{C(L)} \sum_{j \in P_c} \left\langle A_{r_c}^{(c,h)}, \mathbf{x}^{(j)} \right\rangle \left\langle B_{r_{c+1}}^{(c,h)}, \mathbf{x}^{(j)} \right\rangle \right) \left\langle A_{r_{C(L)+1}}^{(0,h)}, \mathbf{x}^{(i)} \right\rangle$$

Now we reorder the above sum by summing over indices of swaps between $P$ and $Q$, i.e. $\beta \in [C]$ such that $P_\beta \neq P_{\beta+1}$, and split the multiplication $\prod_{c=1}^{C(L)}$ according to the crossing indices:

$$= \sum_{h \in [H]^{[C(L)]}} \sum_{r_1, ..., r_{C(L)+1}=1}^{d_a} \sum_{b=0}^{C(L)} \sum_{0=\beta_{b+1}<\beta_b \leq \beta_{b-1}...\leq\beta_1<\beta_0=C(L)} B_{r_1,p}^{(0,h)}$$

$$\left( \left( \prod_{m=0}^{\lfloor \frac{b}{2} \rfloor} \prod_{c=\beta_{2m+1}+1}^{\beta_{2m}} \sum_{j \in P} \left\langle A_{r_c}^{(c,h)}, \mathbf{x}^{(j)} \right\rangle \left\langle B_{r_{c+1}}^{(c,h)}, \mathbf{x}^{(j)} \right\rangle \right) \left\langle A_{r_{C(L)+1}}^{(0,h)}, \mathbf{x}^{(i)} \right\rangle \right)$$

$$\left( \prod_{m=0}^{\lceil \frac{b}{2} \rceil - 1} \prod_{c=\beta_{2m+2}+1}^{\beta_{2m+1}} \sum_{j \in Q} \left\langle A_{r_c}^{(c,h)}, \mathbf{x}^{(j)} \right\rangle \left\langle B_{r_{c+1}}^{(c,h)}, \mathbf{x}^{(j)} \right\rangle \right)$$

Where we assume w.l.o.g that $i \in P$ and therefore $P_{\beta_1}, P_{\beta_1+1}, \ldots, P_{\beta_0-1}, P_{\beta_0} = P$. The above reordering allows pushing the summation of non swapping $r_c$ indices into the $P, Q$ parentheses:

$$= \sum_{h \in [H]^{[C(L)]}} \sum_{b=0}^{C(L)} \sum_{0=\beta_{b+1}<\beta_b \leq \beta_{b-1}...\leq\beta_1\leq\beta_0=C(L)} \sum_{r_{\beta_1+1}, ..., r_{\beta_b+1}=1}^{d_a} B_{r_1,p}^{(0,h)} \qquad (6)$$

$$\left( \left( \underbrace{\sum_{r_{C(L)+1}=1}^{d_a}}_{\substack{\text{just for } \beta_1 < C \\ \text{otherwise ignore}}} \underbrace{\sum_{r_1=1}^{d_a}}_{\substack{\text{used either} \\ \text{in } P \text{ or } Q}} \prod_{m=0}^{\lfloor \frac{b}{2} \rfloor} \sum_{\substack{r_{\beta_{2m+1}+2} \\ \vdots \\ r_{\beta_{2m}}}}^{d_a} \prod_{c=\beta_{2m+1}+1}^{\beta_{2m}} \sum_{j \in P} \left\langle A_{r_c}^{(c,h)}, \mathbf{x}^{(j)} \right\rangle \left\langle B_{r_{c+1}}^{(c,h)}, \mathbf{x}^{(j)} \right\rangle \right) \left\langle A_{r_{C(L)+1}}^{(0,h)}, \mathbf{x}^{(i)} \right\rangle \right)$$

$$\underbrace{\hspace{10cm}}_{\text{function of } P}$$

$$\left( \underbrace{\sum_{r_1=1}^{d_a}}_{\substack{\text{used either} \\ \text{in } P \text{ or } Q}} \prod_{m=0}^{\lceil \frac{b}{2} \rceil - 1} \sum_{r_{\beta_{2m+2}+2}, ..., r_{\beta_{2m+1}}=1}^{d_a} \prod_{c=\beta_{2m+2}+1}^{\beta_{2m+1}} \sum_{j \in Q} \left\langle A_{r_c}^{(c,h)}, \mathbf{x}^{(j)} \right\rangle \left\langle B_{r_{c+1}}^{(c,h)}, \mathbf{x}^{(j)} \right\rangle \right)$$

$$\underbrace{\hspace{10cm}}_{\text{function of } Q}$$

Since the separation rank of each term in the above summation is 1, we proved the following upper bound on the separation rank:

$$sep\left(y_p^{i,L,d_x,H,\Theta},P,Q\right) \leq \sum_{h\in[H]^{[C(L)]}}\sum_{b=0}^{C(L)}\sum_{0=\beta_{b+1}<\beta_b\leq\beta_{b-1}...\leq\beta_1\leq\beta_0=C(L)}\sum_{r_{\beta_1}+1,...,r_{\beta_b}+1=1}^{d_a}1$$

$$= H^{C(L)}\sum_{b=0}^{C(L)}\binom{C(L)}{b}(d_a)^b = H^{C(L)}(d_a+1)^{C(L)} = (H(d_a+1))^{C(L)}$$

We note that unlike the $d_a$ case, the same $H$ index can affect nonconsecutive $M^{(c_1,h)}$, $M^{(c_2,h)}$, therefore we can't simply push the $h$ indices as done for the $r$ indices in eq. (6). □

From here, the upper bound in theorem 1 of the main text follows by

$$\log_3\left(sep\left(y_p^{i,L,d_x,H,\Theta},P,Q\right)\right) \leq \log_3\left((H(d_a+1))^{C(L)}\right) = \frac{3^L-1}{2}\log_3(d_x+H) \quad (7)$$

### A.3 Proof of the upper bound in theorem 2 of the main text

In the following theorem, we show how an upper bound on the separation rank is implied by the polynomial degree of $y_p^{i,L,d_x,H,\Theta}$ in eq. (5). We will use the notation of $\left(\!\!\binom{n}{k}\!\!\right)$ – the multiset coefficient, given in the binomial form by $\binom{n+k-1}{k}$. We will use the identity $|\{a_1\ldots a_n\in\mathbb{Z}\geq 0:\sum_{r=1}^n a_r=k\}| = \left(\!\!\binom{n}{k}\!\!\right)$.

**Theorem 2.** *Defining $C(L):=\frac{3^L-1}{2}$, for any depth $L\geq 1$ input size $N>1$ partition $P\cup Q=[N]$ and output locations $i\in[N]$, $p\in[d_x]$, the following holds:*

$$sep\left(y_p^{i,L,d_x,H,\Theta},P,Q\right) \leq d_x\left(C(L)+1\right)\left(\!\!\binom{d_x}{2C(L)}\!\!\right)\left(\frac{2C(L)}{d_x}+1\right)^{d_x} \quad (8)$$

*Proof.* We begin by opening the inner products in eq. (5), explicitly writing the indices:

$$y_p^{i,L,d_x,H,\Theta} = \sum_{j_1,...,j_{C(L)}=1}^N\sum_{h\in[H]^{[C(L)]}}\sum_{r_1,...,r_{C(L)+1}=1}^{d_a}B_{r_1,p}^{(0,h)}\left(\prod_{c=1}^{C(L)}\left\langle A_{r_c}^{(c,h)},\mathbf{x}^{(j_c)}\right\rangle\left\langle B_{r_{c+1}}^{(c,h)},\mathbf{x}^{(j_c)}\right\rangle\right)\left\langle A_{r_{C(L)+1}}^{(0,h)},\mathbf{x}^{(i)}\right\rangle$$

$$= \sum_{\alpha_1,...,\alpha_{C(L)+1},\beta_1,...,\beta_{C(L)}=1}^{d_x}\sum_{j_1,...,j_{C(L)}=1}^N\sum_{h\in[H]^{[C(L)]}}\sum_{r_1,...,r_{C(L)+1}=1}^{d_a}$$

$$\left(B_{r_1,p}^{(0,h)}A_{r_{C(L)+1},\alpha_{C(L)+1}}^{(0,h)}\mathbf{x}_{\alpha_{C(L)+1}}^{(i)}\prod_{c=1}^{C(L)}A_{r_c,\alpha_c}^{(c,h)}\mathbf{x}_{\alpha_c}^{(j_c)}B_{r_c,\beta_c}^{(c,h)}\mathbf{x}_{\beta_c}^{(j_c)}\right)$$

And separating between coefficients and $\mathbf{x}$'s:

$$= \sum_{\alpha_1,...,\alpha_{C(L)+1},\beta_1,...,\beta_{C(L)}=1}^{d_x}\underbrace{\left(\sum_{h\in[H]^{[C(L)]}}\sum_{r_1,...,r_{C(L)+1}=1}^{d_a}B_{r_1,p}^{(0,h)}A_{r_{C(L)+1},\alpha_{C(L)+1}}^{(0,h)}\prod_{c=1}^{C(L)}A_{r_c,\alpha_c}^{(c,h)}B_{r_c,\beta_c}^{(c,h)}\right)}_{:=\mathcal{T}_{\alpha_1,...,\alpha_{C(L)+1},\beta_1,...,\beta_{C(L)}}}$$

$$\left(\sum_{j_1,...,j_{C(L)}=1}^N\mathbf{x}_{\alpha_{C(L)+1}}^{(i)}\prod_{c=1}^{C(L)}\mathbf{x}_{\alpha_c}^{(j_c)}\mathbf{x}_{\beta_c}^{(j_c)}\right)$$

Now we can group monomials by the powers $n_1,\ldots,n_{d_x}$ of each coordinate

$$= \sum_{\alpha_{C(L)+1}} \underbrace{\sum_{n_1 + \ldots n_{d_x} = 2C(L)}}_{\text{The powers}} \left( \overbrace{\sum_{\substack{\alpha_!, \ldots, \alpha_{C(L)}, \beta_1, \ldots, \beta_{C(L)} \in [d_x] \\ \forall m \in [d_x] \, |\{c \in [C(L)] : \alpha_c = m\}| + |\{c \in [C(L)] : \beta_c = m\}| = n_m}}^{\text{How to distributethe powers between the } c\text{'s}} \mathcal{T}_{\alpha_1, \ldots, \alpha_{C(L)+1}, \beta_1, \ldots, \beta_{C(L)}} \right)$$

$$\chi_{n_1, \ldots, n_{d_x}, \alpha_{C(L)+1}} \left( \mathbf{x}^{(1)}, \ldots, \mathbf{x}^{(N)} \right)$$

Where:

$$\chi_{n_1, \ldots, n_{d_x}, \alpha_{C(L)+1}} \left( \mathbf{x}^{(1)}, \ldots, \mathbf{x}^{(N)} \right) := \underbrace{\sum_{\substack{o_1 + \cdots + o_N = C(L)}}}_{\substack{\text{How many } j \text{ indices} \\ \text{equal to each } [N]}} \underbrace{\sum_{\substack{0 \leq n_{1,1}, \ldots, n_{d_x,N} \leq 2C(L) \\ \forall m \in [d_x] \, \sum_{j=1}^{N} n_{m,j} = n_m \\ \forall j \in [N] \, \sum_{m=1}^{d_x} n_{m,j} = 2o_j}}}_{\text{How to distribute the powers between } [N]} \mathbf{x}_{\alpha_{C(L)+1}}^{(i)} \prod_{j=1}^{N} \prod_{m=1}^{d_x} \left( \mathbf{x}_m^{(j)} \right)^{n_{m,j}}$$

Finally, we need to bound the separation rank of $\chi_{n_1, \ldots, n_{d_x}, \alpha_{C(L)+1}}$. W.l.o.g we choose the partition $P = \left\{ 1, \ldots, \frac{N}{2} \right\}, Q = \left\{ \frac{N}{2} + 1, \ldots, N \right\}$ and $i \in P$ then we can divide the powers between $P, Q$ in the following way:

$$\chi_{n_1, \ldots, n_{d_x}, \alpha_{C(L)+1}} \left( \mathbf{x}^{(1)}, \ldots, \mathbf{x}^{(N)} \right) = \sum_{\substack{0 \leq r_{1,P}, \ldots, r_{d_x,P} \leq 2C(L) \\ 0 \leq r_{1,Q}, \ldots, r_{d_x,Q} \leq 2C(L) \\ \forall m \in [d_x] \, r_{m,P} + r_{m,Q} = n_m}} \sum_{E=0}^{C(L)}$$

$$\underbrace{\left( \sum_{\substack{o_1 + \cdots + o_{\frac{N}{2}} = E}} \sum_{\substack{0 \leq n_{1,1}, \ldots, n_{d_x, \frac{N}{2}} \leq 2C(L) \\ \forall m \in [d_x] \, \sum_{j \in P}^{N} n_{m,j} = r_{m,P} \\ \forall j \in [N] \, \sum_{m=1}^{d_x} n_{m,j} = 2o_j}} \mathbf{x}_{\alpha_{C(L)+1}}^{(i)} \prod_{j \in P} \prod_{m=1}^{d_x} \left( \mathbf{x}_m^{(j)} \right)^{n_{m,j}} \right)}_{\text{function of } P}$$

$$\underbrace{\left( \sum_{\substack{o_{\frac{N}{2}+1} + \cdots + o_N = C(L) - E}} \sum_{\substack{0 \leq n_{1,1}, \ldots, n_{d_x, \frac{N}{2}} \leq 2C(L) \\ \forall m \in [d_x] \, \sum_{j \in Q}^{N} n_{m,j} = r_{m,Q} \\ \forall j \in [N] \, \sum_{m=1}^{d_x} n_{m,j} = 2o_j}} \prod_{j \in Q} \prod_{m=1}^{d_x} \left( \mathbf{x}_m^{(j)} \right)^{n_{m,j}} \right)}_{\text{function of } Q}$$

Thus, since each summand is of separation rank 1, the separation rank of $\chi_{n_1, \ldots, n_{d_x}, \alpha_{C(L)+1}}$ is bounded by the number of summands:

$$(C(L) + 1) \prod_{\beta=1}^{d_x} \left( \binom{2}{r_\beta} \right) \overset{\text{lemma 3}}{\leq} (C(L) + 1) \left( \frac{2C(L)}{d_x} + 1 \right)^{d_x}$$

where the inequality followed from lemma 3. Since we have at most $d_x \left( \binom{d_x}{2C(L)} \right)$ different $\chi$ we conclude that:

$$sep \left( y_p^{i, L, d_x, H, \Theta}, P, Q \right) \leq \underbrace{d_x \left( \binom{d_x}{2C(L)} \right)}_{\text{number of } \chi} (C(L) + 1) \left( \frac{2C(L)}{d_x} + 1 \right)^{d_x}$$

$\square$

From here, the theorem follows by the multiset identity in lemma 4:

$$\log_3\left[sep\left(y_p^{i,L,d_x,H,\Theta},P,Q\right)\right] \le \log_3\left[d_x\left(C\left(L\right)+1\right)\left(\binom{d_x}{2C\left(L\right)}\right)\left(\frac{2C\left(L\right)}{d_x}+1\right)^{d_x}\right] \qquad (9)$$

$$\le \log_3\left[d_x\left(C\left(L\right)+1\right)\left(\frac{2e\left(d_x+2C\left(L\right)\right)}{d_x}\right)^{d_x}\left(\frac{2C\left(L\right)}{d_x}+1\right)^{d_x}\right]$$

$$\le \log_3\left[3^L d_x\left(2e\right)^{d_x}\left(\frac{3^L-1}{d_x}+1\right)^{2d_x}\right]$$

$$\overset{\overbrace{\text{only for } 3^L > d_x}}{\le} \log_3\left[3^L d_x\left(2e\right)^{d_x}\left(2\cdot\frac{3^L-1}{d_x}\right)^{2d_x}\right]$$

$$\le L + \log_3 d_x + d_x\log_3 2e + 2d_x\log_3\left\{\left(\frac{2\cdot 3^L}{d_x}\right)\right\}$$

$$\le \left(2d_x+1\right)L + \log_3 d_x + 2d_x\left(\log_3 2\sqrt{2e}-\log_3 d_x\right) \qquad \square$$

## A.4 The effect of residual connections

(a) Conventional 3-block residual network     (b) Unraveled view of (a)

Figure 1: A residual network in its compressed and unraveled form, taken from Veit et al. [2016].

Having upper-bounded the separation rank of the deep self-attention network defined in section 2.2 of the main text, we comment on the effect of adding residual connections over each layer, as is done in the regular network (described in section 2.1 of the main text). Consider a network composed of a concatenation of the building blocks shown in figure 1(a), taken from Veit et al. [2016]. A building block in layer $l$ includes a module $f_l$, which in our case is the self-attention layer given in eq. (3) of the main text,[1] and a skip connection which adds $f_l$'s input to its output (circles denote addition). Veit et al. [2016] propose an unraveled view of such a network, shown in figure 1(b), which we will employ in the proof of theorem 3 for clarity of presentation.

We begin by proving a lemma that quantifies how the separation rank of the composition of a self-attention layer over a function is related to the function's separation rank:

**Lemma 2.** *Let $g^j \in \mathbb{R}^{d_x}$ be an input vector at position $j$ to a self-attention layer defined by eq. (3) of the main text, and let $K$ be an upper bound to the separation rank of any of the entries $p \in [d_x]$ of any input $g^j \in \mathbb{R}^{d_x}$, i.e., $\forall p \in [d_x], j \in [N] : Sep\left(g_p^j\right) \le K$. Let $y_p^i$ be the $p$th entry of the self-attention layer output at position $i$. Then, an upper bound to the separation rank of $y_p^i \in \mathbb{R}$ is given by:*

$$Sep\left(y_p^i\right) \le \frac{Nd_x^4}{H}K^3$$

*Proof.* Denote by $G \in \mathbb{R}^{d_x \times N}$ the matrix holding $g^j \in \mathbb{R}^{d_x}$ in its $j$th column. Note that by the conditions of the lemma any entry of $G$ upholds $Sep(G_{\alpha\beta}) \leq K$. Writing eq. (3) of the main text as a summation over matrix indices:

$$y_p^i = \sum_{h=1}^{H} \sum_{\alpha_1=1}^{d_x/H} \sum_{\alpha_2=1}^{d_x} \sum_{j=1}^{N} \sum_{\alpha_4=1}^{d_x} \sum_{\alpha_5=1}^{d_x/H} \sum_{\alpha_6=1}^{d_x} W_{p\alpha_1}^{\mathrm{O},h} W_{\alpha_1\alpha_2}^{\mathrm{V},h} G_{\alpha_2 j} (G^\top)_{j\alpha_4} (W^{\mathrm{K},h})_{\alpha_4\alpha_5}^\top W_{\alpha_5\alpha_6}^{\mathrm{Q},h} G_{\alpha_6 i} \quad (10)$$

The lemma follows by multiplying the number of summed terms by an upper bound on the separation rank of each summed term, $K^3$.  □

We now prove a theorem which establishes that the integration of skip connections modifies the upper bound in theorem 1 of the main text by a small factor.

**Theorem 3.** *For $p \in [d_x]$, let $y_p^{i,L}$ be the scalar function computing the $p$th entry of an output vector at position $i \in [N]$ of the depth-$L$ residual network depicted in figure 1, where $f_l$ is the self-attention layer in eq. (3) of the main text. Then:*

$$\log_3 Sep(y_p^{L,i}) \leq L \log_3 L + (4 \log_3 d_x + \log_3 N - \log_3 H) \cdot \frac{3^L - 1}{2}$$

Comparing this dependence to the upper bound in the theorem 1 of the main text, given in eq. (7), this theorem implies that the effect of residual connections is insignificant to our analysis.

*Proof.* Observing figure 1(b) which gives the $L = 3$ example, we upper bound the separation rank of the entire unraveled network by noting that its output is composed from $L + 1$ additions of outputs from branches of depth $l = 0, ...., L$ (0 being the direct link of the input to the output), such that schematically the separation rank at the output of the entire network can be upper bounded by:

$$Sep(y_p^{L,i}) \leq (L+1)Sep(\text{longest branch}(L))$$

where we denoted longest branch$(L)$ as the function at the output of $f_L$, before the addition with the other branches. Noting that the input to $f_L$ can be recursively viewed as an output of an unraveled network of depth $L - 1$, we bound the separation rank of the function at the input to $f_L$ by $L \cdot Sep(\text{longest branch}(L-1))$. Since $f_L$ is a self-attention layer, Lemma 2 implies that $Sep(\text{longest branch}(L)) \leq \frac{Nd_x^4}{H} (L \cdot Sep(\text{longest branch}(L-1)))^3$. Continuing recursively, and inserting the stopping condition $Sep(\text{longest branch}(L = 1)) = \frac{Nd_x^4}{H}$ (since the input to $f_1$ is a specific entry of the input to the entire network, of separation rank 1), we attain:

$$Sep(y_p^{L,i}) \leq \prod_{l=1}^{L} (l+1) \left( \frac{Nd_x^4}{H} \right)^{3^{l-1}},$$

satisfying the theorem.  □

We now prove a theorem which establishes that the integration of skip connections modifies the upper bound in theorem 2 of the main text by a small factor.

**Theorem 4.** *Defining $C(L) := \frac{3^L - 1}{2}$, for $p \in [d_x]$, let $y_{p,residual}^{i,L,d_x,H,\Theta}$ be the scalar function computing the $p$th entry of an output vector at position $i \in [N]$ of the depth-$L$ residual network depicted in figure 1, where $f_l$ is the self-attention layer in eq. (3) of the main text. Then for any partition $P \cup Q = [N]$, the following holds:*

$$sep\left(y_{p,residual}^{i,L,d_x,H,\Theta}, P, Q\right) \leq d_x \left(C(L) + 1\right)^2 \left(\binom{d_x}{2C(L)}\right) \left(\frac{2C(L)}{d_x} + 1\right)^{d_x}$$

Comparing this dependence to the upper bound in the theorem 2 of the main text, given in eq. (8), the above theorem implies that the effect of residual connections is insignificant to our analysis.

*Proof.* We will adapt the proof of theorem 2. All of the arguments remain unchanged, except that we obtain the network structure via lemma 5 instead of lemma 1. Following the new structure we will

have two additional summations, one over $j$ and one over $\alpha$ (see lemma 5), as well as an additional input $X$ factor. We will leave the summation over $j$ during the whole proof, thus multiplying the separation rank by at most $C(L)$. Note that similarity to the $h$ summation, the summation over $\alpha$ has no influence on the separation rank, since it collapses into a single coefficient $\mathcal{T}$. Finally the unput $X$ factor contribute at most $1$ to the separation rank, therefore we can bound the separation rank by $C(L)+1$ times the bound in eq. (8). $\qquad\square$

Finally, since the upper bounds undergo such minor increases in the presence of skip connections, the lower bounds can be left with no further tightening, without affecting the analysis and its conclusions.

## A.5 Technical lemmas

**Lemma 3.** *(inequality of arithmetic and geometric multiset coefficient means)*

*Let $n, k \in \mathbb{N}$ and $\phi : \mathbb{N}^k \to \mathbb{N} := r_1, \ldots r_k \vdash \prod_{j=1}^{k}\left(\binom{n}{r_j}\right)$ then:*

$$\forall r_z, \ldots r_k \in \mathbb{N} \quad \phi(r_1, \ldots r_k) \leq \frac{\left(\prod_{t=1}^{n-1}\left(\frac{M}{k}+t\right)\right)^k}{((n-1)!)^k}$$

*where $M := \sum_{j=1}^{k} r_j$*

*Proof.* Define $f_t := \prod_{j=1}^{k}(r_j + t)$ and $\psi := \prod_{t=1}^{n-1} f_t$ than by the inequality of arithmetic and geometric means

$$\forall t \in [k] \quad f_t \leq \left(\frac{1}{k}\sum_{j=1}^{k}(r_j + t)\right)^k = \left(\frac{M}{k}+t\right)^k$$

Therefore

$$\phi(r_1, \ldots, r_k) = \prod_{j=1}^{k}\left(\binom{n}{r_j}\right) = \prod_{j=1}^{k}\binom{n+r_j-1}{r_j} = \prod_{j=1}^{k}\frac{(n+r_j-1)!}{r_j!(n-1)!}$$

$$= \frac{1}{((n-1)!)^k}\prod_{j=1}^{k}\prod_{t=1}^{n-1}(r_j+t) = \frac{1}{((n-1)!)^k}\prod_{t=1}^{n-1}f_t \leq \frac{\prod_{t=1}^{n-1}\left(\frac{M}{k}+t\right)^k}{((n-1)!)^k}$$

One can see that when $M$ divided by $k$ it hold that

$$\phi\left(\overbrace{\frac{M}{k}, \ldots, \frac{M}{k}}^{k \text{ times}}\right) = \frac{1}{((n-1)!)^k}\prod_{t=1}^{n-1}f_t = \frac{1}{((n-1)!)^k}\prod_{t=1}^{n-1}\left(\frac{M}{k}+t\right)^k = \left(\prod_{t=1}^{n-1}\left(\frac{M}{k}+t\right)\right)^k$$

hence the name of this lemma. $\qquad\square$

**Lemma 4.** $\left(\binom{n}{k}\right) \leq \left(\frac{2e(n+k)}{n}\right)^n$

*Proof.* : by using the inequality $\binom{n}{k} \leq \left(\frac{en}{k}\right)^k$ we have

$$\left(\binom{n}{k}\right) = \binom{n+k-1}{n-1} \leq \left(\frac{2e(n+k)}{n}\right)^n$$

$\qquad\square$

We now prove a lemma which reveals the alternation to the network structure as expressed in eq. (2) when taking skip connections into account.

**Lemma 5.** *Defining $C(L) := \frac{3^L - 1}{2}$, any depth $L$ skip connection composition of the self-attention layers defined in eq. 3 of the main text can be written as:*

$$Y^{L,d_x,H,\Theta} = X + \sum_{j=1}^{C(L)} \sum_{\alpha=1}^{n_j} \sum_{h \in [H]^{[j]}} B^{(0,h,j,\alpha)T} M^{(1,h,j,\alpha)} \cdots M^{(j,h,j,\alpha)} A^{(0,h,j,\alpha)} X \qquad (11)$$

*where $\forall j \in [C(L)] \quad n_j \geq 0$ and $\forall \alpha \in [n_j] \ h \in [H]^{[j]} \ 0 \leq c \leq j : M^{(c,h)} = A^{(c,h)} X X^T B^{(c,h)T}$ and $A^{(c,h)}, B^{(c,h)} \in \mathbb{R}^{d_a \times d_x}$.*

*Proof.* By Induction on $L$. Base case:

$$Y^{(1)}(X) = X + \sum_{h=1}^{H} \underbrace{W^{O,h}}_{B^T} \underbrace{W^{V,h} X X^T (W^{K,h})^T}_{M} \underbrace{W^{Q,h}}_{A} X$$

$$Y^{(L+1)}(X) = \sum_{h=1}^{H} W^{O,h} W^{V,h} Y^{(L)}(X) Y^{(L)}(X)^T (W^{K,h})^T W^{Q,h} Y^{(L)}(X)$$

Now, rewriting $Y^{(L)}$ as $X + (Y^{(L)} - X)$ yields:

$$Y^{(L+1)}(X) = \sum_{E,F,G \in \{X, Y^{(L)} - X\}} \sum_{h=1}^{H} W^{O,h} W^{V,h} E F^T (W^{K,h})^T W^{Q,h} G$$

Now, substituting in the induction hypothesis on the structure of $Y^{(L)}(X)$ yields:

$$Y^{(L+1)}(X) = \sum_{E,F,G \in \left\{ X, \sum_{j=1}^{C(L)} \sum_{\alpha=1}^{n_j} \sum_{h \in [H]^{[j]}} B^{(0,h,j,\alpha)T} M^{(1,h,j,\alpha)} \cdots M^{(j,h,j,\alpha)} A^{(0,h,j,\alpha)} X \right\}}$$

$$\sum_{h=1}^{H} W^{O,h} W^{V,h} E F^T (W^{K,h})^T W^{Q,h} G$$

Similarly to eq. (3) each of the 8 terms in the outer summation is of the required form, thus we complete the proof. $\qquad \square$

## B  Lower bounds on the separation rank

### B.1  preliminaries

#### B.1.1  Tensors and their matricization

We begin by laying out basic concepts in tensor theory required for the upcoming analysis. The core concept of a *tensor* may be thought of as a multi-dimensional array. The *order* of a tensor is defined to be the number of indexing entries in the array, referred to as *modes*. The *dimension* of a tensor in a particular mode is defined as the number of values taken by the index in that mode. If $\mathcal{A}$ is a tensor of order $N$ and dimension $M_i$ in each mode $i \in [N]$, its entries are denoted $\mathcal{A}_{d_1 \ldots d_N}$, where the index in each mode takes values $d_i \in [M_i]$.

We will make use of the concept of the *matricization of $\mathcal{A}$ w.r.t. the balanced partition* $(I, J)$, denoted $[\![\mathcal{A}]\!]_{I,J} \in \mathbb{R}^{M^{N/2} \times M^{N/2}}$, which is essentially the arrangement of the tensor elements as a matrix whose rows correspond to $I$ and columns to $J$. Suppose $\mathcal{A} \in \mathbb{R}^{M \times \cdots \times M}$ is a tensor of order $N$, and let $(I, J)$ be a balanced partition of $[N]$, *i.e.* $I$ and $J$ are disjoint size $N/2$ subsets of $[N]$ whose union gives $[N]$. The *matricization of $\mathcal{A}$ w.r.t. the partition* $(I, J)$, denoted $[\![\mathcal{A}]\!]_{I,J}$, is the $M^{N/2}$-by-$M^{N/2}$ matrix holding the entries of $\mathcal{A}$ such that $\mathcal{A}_{d_1 \ldots d_N}$ is placed in row index $1 + \sum_{t=1}^{N/2} (d_{i_t} - 1) M^{N/2-t}$ and column index $1 + \sum_{t=1}^{N/2} (d_{j_t} - 1) M^{N/2-t}$.

### B.1.2 Grid tensors provide lower bounds for the separation rank

We now present the concept of grid tensors, which are a form of function discretization [Hackbusch, 2012]. Essentially, the function is evaluated for a set of points on an exponentially large grid in the input space and the outcomes are stored in a tensor. Formally, fixing a set of *template* vectors $\mathbf{x}^{(1)}, \ldots, \mathbf{x}^{(M)} \in \mathbb{R}^{d_x}$, the points on the grid are the set $\{(\mathbf{x}^{(d_1)}, \ldots, \mathbf{x}^{(d_N)})\}_{d_1, \ldots, d_N=1}^{M}$. Given a function $y(\mathbf{x}^1, \ldots, \mathbf{x}^N)$, the set of its values on the grid arranged in the form of a tensor are called the grid tensor induced by $y$, denoted $\mathcal{A}(y)_{d_1, \ldots, d_N} \equiv y(\mathbf{x}^1 = \mathbf{x}^{(d_1)}, \ldots, \mathbf{x}^N = \mathbf{x}^{(d_N)})$.

The following claim establishes a fundamental relation between a function's separation rank (see section 3 of the main text) and the rank of the matrix obtained by the corresponding grid tensor matricization. This relation, which holds for all functions, is formulated below for functions realized by self-attention networks:

**Claim 1.** *For $p \in [d_x]$, let $y_p^{i,L,d_x,H,\Theta}$ be the scalar function computing the pth entry of an output vector at position $i \in [N]$ of the depth-L self-attention network with hidden dimension $d_x$ and $H$ attention heads per layer, defined in eqs. 3 and 4 of the main text. Then, for any integer $M$ and any set of template vectors $\mathbf{x}^{(1)}, \ldots, \mathbf{x}^{(M)} \in \mathbb{R}^{d_x}$ it holds that:*

$$\mathrm{sep}_{(I,J)}\left(y_p^{i,L,d_x,H,\Theta}\right) \geq \mathrm{rank}\left([\![\mathcal{A}(y_p^{i,L,d_x,H,\Theta})]\!]_{I,J}\right), \tag{12}$$

*where $\mathcal{A}(y_p^{i,L,d_x,H,\Theta})$ is the grid tensor of $y_p^{i,L,d_x,H,\Theta}$ with respect to the above template vectors.*

*Proof.* If $\mathrm{sep}_{(I,J)}\left(y_p^{i,L,d_x,H,\Theta}\right) = \infty$ then the inequality is trivially satisfied. Otherwise, assume that $\mathrm{sep}_{(I,J)}\left(y_p^{i,L,d_x,H,\Theta}\right) = K \in \mathbb{N}$, and let $\{g_\nu^I, g_\nu^J\}_{\nu=1}^{K}$ be the functions of the respective decomposition to a sum of separable functions, i.e. that the following holds:

$$y_p^{i,L,d_x,H,\Theta}(\mathbf{x}^1, \ldots, \mathbf{x}^N) = \sum_{\nu=1}^{K} g_\nu^I(\mathbf{x}^j : j \in I) \cdot g_\nu^J(\mathbf{x}^j : j \in J).$$

Then, by definition of the grid tensor, for any template vectors $\mathbf{x}^{(1)}, \ldots, \mathbf{x}^{(M)} \in \mathbb{R}^{d_x}$ the following equality holds:

$$\begin{aligned}
\mathcal{A}(y_p^{i,L,d_x,H,\Theta})_{d_1, \ldots, d_N} &= \sum_{\nu=1}^{K} g_\nu^I(\mathbf{x}^{(d_j)} : j \in I) \cdot g_\nu^J(\mathbf{x}^{(d_j)} : j \in J) \\
&\equiv \sum_{\nu=1}^{K} V_{d_j:j\in[I]}^{\nu} U_{d_j:j\in[J]}^{\nu},
\end{aligned}$$

where $V^\nu$ and $U^\nu$ are the tensors holding the values of $g_\nu^I$ and $g_\nu^J$, respectively, at the points defined by the template vectors. Under the matricization according to the $(I, J)$ partition, it holds that $[\![V^\nu]\!]_{I,J}$ and $[\![U^\nu]\!]_{I,J}$ are column and row vectors, respectively, which we denote by $\mathbf{v}_\nu$ and $\mathbf{u}_\nu^T$. It follows that the matricization of the grid tensor is given by:

$$[\![\mathcal{A}(y_p^{i,L,d_x,H,\Theta})]\!]_{I,J} = \sum_{\nu=1}^{K} \mathbf{v}_\nu \mathbf{u}_\nu^T,$$

which means that $\mathrm{rank}\left([\![\mathcal{A}(y_p^{i,L,d_x,H,\Theta})]\!]_{I,J}\right) \leq K = \mathrm{sep}_{(I,J)}\left(y_p^{i,L,d_x,H,\Theta}\right)$. $\qquad\square$

### B.1.3 Method for bounding the grid tensor's rank

Claim 1 assures us that the separation rank of the function realized by a self-attention network is lower bounded by the rank of the matrix obtained by the corresponding grid tensor matricization, for any choice of template vectors. Specifically:

$$\mathrm{sep}_{(I,J)}\left(y_p^{i,L,d_x,H,\Theta}\right) \geq \mathrm{rank}\left([\![\mathcal{A}(y_p^{i,L,d_x,H,\Theta})]\!]_{I,J}\right).$$

Thus, proving that $\mathrm{rank}\left([\![\mathcal{A}(y_p^{i,L,d_x,H,\Theta})]\!]_{I,J}\right)$ is higher than the lower bounds stated in theorems 1 and 2 of the main text for all of the values of the parameters $\Theta$ but a set of Lebesgue measure zero, would satisfy the theorems.

We note that since the network's operation is polynomial in $\Theta$, then the entries of the grid tensor are also polynomial. Sharir et al. [2016] prove a claim regarding the prevalence of the maximal matrix rank for matrices whose entries are polynomial functions. Essentially, they show that it suffices to find a single configuration of the parameters, denoted $\theta \in \mathbb{R}^K$ (where $K$ is the number of scalar parameters), for which the resultant matrix is of rank $r$, in order to show the rank is at least $r$ for all configurations in $\mathbb{R}^K$ but a set of measure zero in $\mathbb{R}^K$. For simplicity of the proof we will find a single configuration $\theta \in \mathbb{C}^K$ for which the resultant matrix is of the required rank. We therefore modify the original claim to fit this setting, still proving the rank is lower bounded for all configurations in $\mathbb{R}^K$ but a set of measure zero in $\mathbb{R}^K$:

**Claim 2.** *Let $M, N, K \in \mathbb{N}$, $1 \leq r \leq \min\{M, N\}$ and an $M \times N$ matrix $A$ where each entry is a polynomial mapping $A_{ij}$ over $K$ variables for every $i \in [M]$ and $j \in [N]$. If there exists a point $\theta \in \mathbb{F}^K$, where $\mathbb{F}$ is either $\mathbb{R}$ or $\mathbb{C}$, s.t. $rank(A(\theta)) \geq r$, then the set $\{\theta \in \mathbb{R}^K : rank(A(\theta)) < r\}$ has zero measure (w.r.t. the Lebesgue measure over $\mathbb{R}^K$).*

*Proof.* (based on a proof in Sharir et al. [2016]) Recall that $\text{rank}(A(\theta)) \geq r$ iff there exits a non-zero $r \times r$ minor of $A(\theta)$. Note that a minor of $A(\theta)$ is polynomial in the entries of $A(\theta)$, and so it is polynomial in $\theta$ as well. Let $c = \binom{M}{r} \cdot \binom{N}{r}$ be the number of minors in $A$, denote the minors by $\{f_i(\theta)\}_{i=1}^c$, and define a new polynomial function $f(\theta) = \sum_{i=1}^c f_i(\theta)^2$. It thus holds that $f(\theta) = 0$ iff for all $i \in [c]$ it holds that $f_i(\theta) = 0$, i.e. $f(\theta) = 0$ iff $\text{rank}(A(\theta)) < r$.

Now, $f(\theta)$ is a polynomial in the entries of $\theta$, and so it either vanishes on a set of zero measure in $\mathbb{R}^K$, or it is the zero polynomial (see Caron and Traynor [2005] for proof). Since we assumed that there exists $\theta \in \mathbb{F}^K$ s.t. $\text{rank}(A(\theta)) \geq r$, the latter option is not possible. $\square$

### B.2   Proof of the lower bounds in theorems 1 and 2 of the main text

In this section, we show there exists an assignment for the weight matrices of a self-attention network, along with a specific choice of template vectors, for which $\text{rank}\left(\llbracket \mathcal{A}(y_p^{i,L,d_x,H,\Theta}) \rrbracket_{I,J}\right)$ surpasses the lower bounds stated in theorems 1 and 2 of the main text in the appropriate depth to width ratios. In accordance with Claim 2, the lower bounds in the theorems will follow since such an assignment implies this rank is achieved for all configurations of the self-attention network weights but a set of Lebesgue measure zero.

*Proof.* (of lower bounds in theorems 1 and 2 of the main text).

Relying on claim 1 we will bound the separation rank from below via the rank of the matricization w.r.t. a partition $(I, J)$ of a grid tensor induced by $y_p^{i,L,d_x,H,\Theta}$, computed by any set of template vectors: $\text{sep}_{(I,J)}\left(y_p^{i,L,d_x,H,\Theta}\right) \geq \text{rank}\left(\llbracket \mathcal{A}(y_p^{i,L,d_x,H,\Theta}) \rrbracket_{I,J}\right)$. Relying on claim 2, we ensure that the rank of $\llbracket \mathcal{A}(y_p^{i,L,d_x,H,\Theta}) \rrbracket_{I,J}$ is above a certain value almost everywhere by finding an assignment of the network parameters for which it achieves this value.

Lemma 6 assures us that for any matrix $V \in \mathbb{R}^{M/2 \times (d_x - H)/2}$ with $l^2$ normalized rows, there exists a choice of $M + 1$ template vectors $\mathbf{x}^{(1)}, \ldots, \mathbf{x}^{(M+1)} \in \mathbb{R}^{d_x}$, as well as an assignment to the self-attention network weights for which:

$$\llbracket \mathcal{A}(y_p^{i,L,d_x,H,\Theta}) \rrbracket_{\tilde{I},\tilde{J}} = \text{Const.} \cdot \left(VV^T\right)^{\odot(3^{L-2})}, \tag{13}$$

where $\llbracket \mathcal{A}(y_p^{i,L,d_x,H,\Theta}) \rrbracket_{\tilde{I},\tilde{J}}$ is a sub-matrix of the grid tensor matricization $\llbracket \mathcal{A}(y_p^{i,L,d_x,H,\Theta}) \rrbracket_{I,J}$ of size $M/2 \times M/2$ and $\odot$ represents the Hadamard power operation, i.e., $\left(A^{\odot k}\right)_{ij} = A_{ij}^k$. Since proving the existence of a sub-matrix of a certain rank lower-bounds the rank of the full matrix by this rank, it suffices to find a matrix $V$ such that $\text{rank}\left(\left(VV^T\right)^{\odot(3^{L-2})}\right)$ upholds the stated dependence.

Noting that the operation of raising a rank $r$ matrix to the Hadamard power of $p$ results in a matrix upper bounded by $\left(\!\!\binom{r}{p}\!\!\right)$ (see proof in Amini et al. [2012] for example) with the notation of the multiset coefficient $\left(\!\!\binom{n}{k}\!\!\right) := \binom{n+k-1}{k}$, and that the rank of $VV^T$ is upper bounded by $(d_x - H)/2$, we choose the dimension $M/2 = \left(\!\!\binom{(d_x-H)/2}{3^{L-2}}\!\!\right)$ to facilitate the rank increase.

For this choice, observe that it suffices to prove that the sub-matrix $[\![\mathcal{A}(y_p^{i,L,d_x,H,\Theta})]\!]_{\tilde{I},\tilde{J}} \in R^{M/2 \times M/2}$ is fully ranked in order to satisfy the theorems. This follows by using the identity $\binom{n}{k} \geq \left(\frac{n}{k}\right)^k$ we have: $\left(\!\!\binom{n}{k}\!\!\right) = \binom{n+k-1}{k} = \binom{n+k-1}{n-1} \geq \max\left\{\left(\frac{n-1}{k}+1\right)^k, \left(\frac{k}{n-1}+1\right)^{n-1}\right\}$

And accordingly:

$$\left(\!\!\binom{(d_x-H)/2}{3^{L-2}}\!\!\right) \geq \max\left\{\left(\frac{(d_x-H)/2-1}{3^{L-2}}+1\right)^{3^{L-2}}, \left(\frac{3^{L-2}}{(d_x-H)/2-1}+1\right)^{(d_x-H)/2-1}\right\}$$

and the log of this bounds the expressions in the theorems' lower bounds, where for each regime the tighter lower bound is used.

Defining for brevity $d := {(d_x-H)}/2$ and $\lambda := 3^{L-2}$, it remains only to find a specific matrix $V \in \mathbb{R}^{\binom{d}{\lambda} \times d}$ with $l^2$ normalized rows such that the operation of taking the rank $d$ matrix $VV^\top$ to the Hadamard power of $\lambda$ would result in a fully ranked matrix. We will provide such a matrix, and prove for it that:

$$\left(VV^\top\right)^{\odot\lambda} = \sum_{k=1}^{\binom{d}{\lambda}} \mathbf{a}^{(k)} \otimes \mathbf{b}^{(k)} \tag{14}$$

for $\{\mathbf{a}^{(k)}\}_{k=1}^{\binom{d}{\lambda}}$ and $\{\mathbf{b}^{(k)}\}_{k=1}^{\binom{d}{\lambda}}$ which are two sets of linearly independent vectors.

For $\alpha, \beta \in [\left(\!\!\binom{d}{\lambda}\!\!\right)]$, observing an entry of $\left(VV^\top\right)^{\odot\lambda}$:

$$\left(\left(VV^\top\right)^{\odot\lambda}\right)_{\alpha\beta} = \left(VV^\top\right)_{\alpha\beta}^\lambda = \left(\sum_{r=1}^d v_r^{(\alpha)} v_r^{(\beta)}\right)^\lambda = \tag{15}$$

$$\sum_{k_1+\cdots+k_d=\lambda} \binom{\lambda}{k_1,\ldots,k_d}\left[\prod_{r=1}^d \left(v_r^{(\alpha)}\right)^{k_r}\right]\left[\left[\prod_{r=1}^d \left(v_r^{(\beta)}\right)^{k_r}\right]\right] \tag{16}$$

where the first equality follows from the definition of the Hadamard power, in the section we denoted $v_r^{(\alpha)}, v_r^{(\beta)}$ as the $r$th entries in rows $\alpha$ and $\beta$ of $V$, and in the second line we expanded the power with the multinomial identity. Identifying the form of eq. (16) with the schematic form of eq. (14), it remains to find a specific matrix $V \in \mathbb{R}^{\binom{d}{\lambda} \times d}$ with $l^2$ normalized rows for which the size $\left(\!\!\binom{d}{\lambda}\!\!\right)$ set $\left\{\mathbf{a}^{(k_1,\ldots,k_d)}\right\}_{k_1+\cdots+k_d=\lambda}$ is linearly independent, where $a_\alpha^{(k_1,\ldots,k_d)} = \prod_{r=1}^d \left(v_r^{(\alpha)}\right)^{k_r}$.

We show this is the case for $V$ in which the rows are each associated with one of $\left(\!\!\binom{d}{\lambda}\!\!\right)$ configurations of distributing $d$ integer numbers that sum up to $\lambda$, *i.e.*, in which each row is associated with specific $\left\{q_1^\alpha, \ldots, q_d^\alpha \geq 0, \sum_{r=1}^d q_r^\alpha = \lambda\right\}$. Explicitly, we take the rows $\mathbf{v}_r^{(\alpha)}$ to be:

$$\forall r \in [d] : v_r^{(\alpha)} = \Omega^{q_r^\alpha}\Big/\sqrt{\sum_{r'=1}^d \Omega^{2q_{r'}^\alpha}}$$

Given this $V$, each vector in the above defined set $\left\{\mathbf{a}^{(k_1,\ldots,k_d)}\right\}_{k_1+\cdots+k_d=\lambda}$ is equal to:

$$a_\alpha^{(k_1,\ldots,k_d)} = \prod_{r=1}^d \left(v_r^{(\alpha)}\right)^{k_r} = \prod_{r=1}^d \left(\frac{\Omega^{q_r^\alpha}}{\sqrt{\sum_{r'=1}^d \Omega^{2q_{r'}^\alpha}}}\right)^{k_r} = \frac{\prod_{r=1}^d \Omega^{q_r^\alpha k_r}}{\prod_{r=1}^d \left(\sum_{r'=1}^d \Omega^{2q_{r'}^\alpha}\right)^{\frac{k_r}{2}}}$$

$$= \left(\sum_{r'=1}^d \Omega^{2q_{r'}^\alpha}\right)^{-\frac{\lambda}{2}} \cdot \left[\Omega^{\sum_{r=1}^d q_r^\alpha k_r}\right]$$

Observing that the factor attained from the normalization depends only on the rows and doesn't vary with the different vectors labeled by $(k_1, \ldots, k_d)$, we note it does not affect their linear dependence

(amounts to a multiplication by a diagonal matrix with non-zero entries on the diagonal - does not affect the rank).

We prove that the set $\left\{\hat{\mathbf{a}}^{(k_1,\ldots,k_d)}\right\}_{k_1+\cdots+k_d=\lambda}$ for $\hat{a}_\alpha^{(k_1,\ldots,k_d)} = \Omega^{\sum_{r=1}^d q_r^\alpha k_r}$ is linearly independent by arranging it as the columns of the matrix $A \in \mathbb{R}^{\binom{d}{\lambda} \times \binom{d}{\lambda}}$, and showing that $A$ is fully ranked.

Since the elements of $A$ are polynomial in $\Omega$, then as lemma 7 shows, it is sufficient to show that there exists a single contributor to the determinant of $A$ that has the highest degree of $\Omega$ in order to ensure that the matrix is fully ranked for all values of $\Omega$ but a finite set, so $\Omega$ should simply be chosen to be any number that is outside of this set. Observing the summands of the determinant, i.e. $\Omega^{\sum_{q_1+\cdots+q_d=\lambda}\langle \mathbf{q}, \sigma(\mathbf{q})\rangle}$, where $\sigma$ is a permutation on the columns of $A$, lemma 8 assures us the existence of a strictly maximal contributor, satisfying the conditions of lemma 7, thus the set $\left\{\hat{\mathbf{a}}^{(k_1,\ldots,k_d)}\right\}_{k_1+\cdots+k_d=\lambda}$ is linearly independent, and the lower bounds in the theorems follow.  $\square$

## B.3  Technical lemmas

The following lemma details the assignment of the self-attention network weights and the choice of template vectors which help us establish the theorems.

**Lemma 6.** *For any balanced partition of $[N]$, denoted $(I, J)$, for any even $M$, and for any matrix $V \in \mathbb{R}^{M/2 \times (d_x - H)/2}$ with rows that are $l^2$ normalized, there exists a choice of $M + 1$ template vectors $\mathbf{x}^{(1)}, \ldots, \mathbf{x}^{(M+1)} \in \mathbb{R}^{d_x}$, as well as an assignment to the self-attention network weights, for which:*

$$\llbracket \mathcal{A}(y_p^{i,L,d_x,H,\Theta}) \rrbracket_{\tilde{I},\tilde{J}} = Const. \cdot \left(VV^T\right)^{\odot 3^{L-2}}, \tag{17}$$

*where $\llbracket \mathcal{A}(y_p^{i,L,d_x,H,\Theta}) \rrbracket_{\tilde{I},\tilde{J}}$ is a sub-matrix of the grid tensor matricization $\llbracket \mathcal{A}(y_p^{i,L,d_x,H,\Theta}) \rrbracket_{I,J}$ of size $M/2 \times M/2$ and $\odot$ represents the Hadamard power operation, i.e., $\left(A^{\odot k}\right)_{ij} = A_{ij}^k$.*

*Proof.* We present below a choice of weights and template vectors that yields the stated form for a sub-matrix of $\llbracket \mathcal{A}(y_p^{i,L,d_x,H,\Theta}) \rrbracket_{I,J}$. Subsequently we will plug these values into the self-attention operation stated in eq. 3 of the main text, and prove that this form follows.

Though the proof has many technical details, it has 3 essential parts. We first choose the weights of the first layer so that the outputs in all locations are the same and equal to a summation of the input vectors. Because the weight matrices are not $d_x \times d_x$ but are decomposed through the attention dimension $d_a \times d_x$ or $d_x \times d_a$, then we divide the coordinates of the $d_x$-dimensional vectors into contiguous segments of length $d_a$, and set the weights to either project these segments to the $d_a$-dimensional space or invert this mapping with added zero-padding. For the second part, we set the key and query matrices to use the same "projections" we used in the first layer to compute inner-products between each segment, while setting the value and output matrices to preserve each head's segment (with zero-padded coordinates). For the remainder of the network's layers, we use the previous step to compute increasingly larger powers of the norm of the vector computed in the first layer, by reconstructing the squared-norm from the inner products of each segment. The template vectors (and parameters) are chosen such that the square of this norm will be equal to $VV^T$.

The assignment to the network weights:

$$
W_{i,j}^{V,1,h} = \frac{1}{N} \cdot
\begin{cases}
1_{i=j-d_a \cdot (h-1)} & \begin{array}{c} d_a(h-1) < j \le d_a(h-1) + \frac{d_a-1}{2} \\ 0 < i \le \frac{d_a-1}{2} \end{array} \\[2ex]
\mathbf{i} \cdot 1_{i=j-d_a \cdot (h-1)-\frac{d_a-1}{2}} & \begin{array}{c} d_a(h-1) + \frac{d_a-1}{2} < j \le d_a h - 1 \\ 0 < i \le \frac{d_a-1}{2} \end{array} \\[2ex]
-1_{i=j-d_a \cdot (h-1)} & \begin{array}{c} d_a(h-1) < j \le d_a(h-1) + \frac{d_a-1}{2} \\ \frac{d_a-1}{2} < i \le d_a - 1 \end{array} \\[2ex]
-\mathbf{i} \cdot 1_{i=j-d_a \cdot (h-1)-\frac{d_a-1}{2}} & \begin{array}{c} d_a(h-1) + \frac{d_a-1}{2} < j \le d_a h - 1 \\ \frac{d_a-1}{2} < i \le d_a - 1 \end{array} \\[2ex]
1 & j = d_a h, \frac{d_a-1}{2} < i \le d_a \\[1ex]
0 & \text{Otherwise}
\end{cases}
$$

$$
W_{i,j}^{O,l,h} =
\begin{cases}
1_{j=i-d_a(h-1)} & d_a(h-1) < i \le d_a h \\
0 & \text{Otherwise}
\end{cases}
$$

$$
\forall 1<l<L, W_{i,j}^{V,l,h} =
\begin{cases}
1_{i=j-d_a \cdot (h-1)} & d_a(h-1) < j \le d_a h \\
0 & \text{Otherwise}
\end{cases}
$$

$$
W_{i,j}^{V,L,h} = \mathbf{i} \cdot 1_{j=d_a}
$$

$$
W_{i,j}^{K,1,h} = W_{i,j}^{Q,1,h} = 1_{i=1 \wedge j=d_a}
$$

$$
W_{i,j}^{K,2,h} = W_{i,j}^{Q,2,h} =
\begin{cases}
1_{i=j-d_a \cdot (h-1)} & \begin{array}{c} d_a(h-1) < j \le d_a(h-1) + \frac{d_a-1}{2} \\ 0 < i \le \frac{d_a-1}{2} \end{array} \\[2ex]
0 & \text{Otherwise}
\end{cases}
$$

$$
\forall l>2, W_{i,j}^{K,l,h} = W_{i,j}^{Q,l,h} =
\begin{cases}
1 & i = 1 \wedge j \bmod d_a \ne 0 \\
0 & \text{Otherwise}
\end{cases}
$$

In the above, we denoted the complex root of $-1$ as $\mathbf{i}$, to differentiate it from the index $i$. The choice of template vectors:

$$
x_j^{(i)} =
\begin{cases}
V_{i,\phi(j)} & i \le M/2 \wedge (j-1) \bmod d_a < \frac{d_a-1}{2} \\
V_{i-M/2+1,\phi\left(j-\frac{d_a-1}{2}\right)} & \frac{M}{2} < i \le M \wedge \frac{d_a-1}{2} \le (j-1) \bmod d_a < d_a - 1 \\
1 & (j-1) \bmod d_a = d_a - 1 \\
0 & \text{Otherwise}
\end{cases}
$$

where $\phi(j) \equiv \lfloor j-1/d_a \rfloor \cdot (d_a - 1) + (j-1 \bmod d_a) + 1$.

W.l.o.g. we can assume that $I = \{1, \ldots, N/2\}, J = \{N/2 + 1, \ldots, N\}$. We examine the sub-matrix defined by the following indices:

$$
\tilde{I} = \{(i_1, \ldots, i_{N/2}) : 1 \le i_1 \le M/2 \wedge \forall k > 1, i_k = M + 1\} \tag{18}
$$

$$
\tilde{J} = \{(j_1, \ldots, j_{N/2}) : M/2 < j_1 \le M \wedge \forall k > 1, j_k = M + 1\} \tag{19}
$$

With all of the above in place, we are ready to prove that the resulting sub-matrix has the form of eq. (17). We begin with the output of the first self-attention layer:

$$\mathbf{y}^{(1,i)}(\mathbf{x}^{(d_1)},\ldots,\mathbf{x}^{(d_N)})_k = \sum_{j=1}^{N}\sum_{h=1}^{H}\left\langle W^{Q,1,h}\mathbf{x}^{(d_i)}, W^{K,1,h}\mathbf{x}^{(d_j)}\right\rangle (W^{O,1,h}W^{V,1,h}\mathbf{x}^{(d_j)})_k \quad (20)$$

$$\overset{1}{=} \sum_{j=1}^{N}\sum_{h=1}^{H}\overbrace{x_{d_a}^{(d_i)}}^{=1}\cdot\overbrace{x_{d_a}^{(d_j)}}^{=1}(W^{O,1,h}W^{V,1,h}\mathbf{x}^{(d_j)})_k \quad (21)$$

$$\overset{2}{=} \left(\left(\sum_{h=1}^{H}W^{O,1,h}W^{V,1,h}\right)\left(\mathbf{x}^{(i_1)}+\mathbf{x}^{(j_1)}+(N-2)\mathbf{x}^{(M+1)}\right)\right)_k \quad (22)$$

$$\overset{3}{=}\begin{cases}1 & (k-1)\bmod d_a = d_a-1 \\ V_{i_1,\phi(k)}+\mathbf{i}V_{j_1,\phi(k)} & (k-1)\bmod d_a < \frac{d_a-1}{2} \\ 1-V_{i_1,\phi(k-\frac{d_a-1}{2})}-\mathbf{i}V_{j_1,\phi(k-\frac{d_a-1}{2})} & \text{Otherwise}\end{cases} \quad (23)$$

where (1) is because $W^{Q,1,h}=W^{K,1,h}$ are matrices that are zero everywhere except for entry $(1,d_a)$, (2) because when summing over the locations, only $i_1$ and $j_1$ are different from $M+1$, and (3) because applying the value and output matrices on any template vector $\mathbf{u}$ results in:

$$\left(W^{O,1,h}W^{V,1,h}\mathbf{u}\right)_k = \sum_{\alpha=1}^{d_a}W_{k,\alpha}^{O,1,h}\sum_{\beta=1}^{d_x}W_{\alpha,\beta}^{V,1,h}u_\beta \quad (24)$$

$$= \sum_{\alpha=1}^{d_a}W_{k,\alpha}^{O,1,h}\overbrace{\begin{cases}u_{d_ah+\alpha-1}+\mathbf{i}\cdot u_{d_ah+\alpha-1+\frac{d_a-1}{2}} & \alpha\leq\frac{d_a-1}{2} \\ \frac{1}{N}-u_{d_ah+\alpha-1}-\mathbf{i}\cdot u_{d_ah+\alpha-1+\frac{d_a-1}{2}} & \frac{d_a-1}{2}<\alpha\leq d_a-1 \\ \frac{1}{N} & \text{Otherwise}\end{cases}}^{\equiv\hat{u}_\alpha} \quad (25)$$

$$= \begin{cases}\hat{u}_{((k-1)\bmod d_a)+1} & d_a(h-1)\leq k < d_ah \\ 0\end{cases} \quad (26)$$

At this point, notice that for any $i\in[N]$, $\mathbf{y}^{(1,i)}$ is the same, and we denote it with $\mathbf{v}$. Note that it is a vector composed of $H$ $d_a$-dimensional sub-vectors, each composed of a $\frac{d_a-1}{2}$-dimensional sub-vector and its complement in the next $\frac{d_a-1}{2}$ indices, followed by a fixed value of 1.

Next, we will compute the result of the second layer, where we use the fact that every position is equal to $\mathbf{v}$ to drop the reference to a specific location $i$, i.e., $\mathbf{y}^{(l,i)}=\mathbf{y}^{(l)}$:

$$\mathbf{y}_k^{(2)} = N\sum_{h=1}^{H}\left\langle W^{Q,2,h}\mathbf{v}, W^{K,2,h}\mathbf{v}\right\rangle (W^{O,2,h}W^{V,2,h}\mathbf{v})_k \quad (27)$$

$$= N\sum_{h=1}^{H}\left\langle\tilde{\mathbf{v}}^{(h)},\tilde{\mathbf{v}}^{(h)}\right\rangle\mathbf{v}^{(h)}, \quad (28)$$

where we used the notation $v_k^{(h)} = v_k\cdot\mathbf{1}_{d_a(h-1)\leq k<d_ah}$, i.e., a vector that is equal to $v_k$ on the $h$'th $d_a$-dimensional segment and otherwise filled with zeros, as well as the notation $\tilde{v}_k^{(h)} = v_k\cdot\mathbf{1}_{d_a(h-1)\leq k\leq d_a(h-1)+\frac{d_a-1}{2}}$. The last equality is because all matrices in this layer essentially just project the $d_a$-dimensional sub-vector of $\mathbf{v}$ for its respective head $h$.

For the third layer we get:

$$\mathbf{y}^{(3)} = N \sum_{h=1}^{H} \left\langle W^{Q,2,h}\mathbf{y}^{(2)}, W^{K,2,h}\mathbf{y}^{(2)} \right\rangle (W^{O,2,h}W^{V,2,h}\mathbf{y}^{(2)}) \tag{29}$$

$$\overset{1}{=} N \sum_{h=1}^{H} \left( \sum_{r \bmod d_a \neq 0} y_r^{(2)} \right)^2 \mathbf{y}^{(2),h} \tag{30}$$

$$\overset{2}{=} N \sum_{h=1}^{H} \left( N \sum_{h'=1}^{H} \left\langle \tilde{\mathbf{v}}^{(h')}, \tilde{\mathbf{v}}^{(h')} \right\rangle \right)^2 N \left\langle \tilde{\mathbf{v}}^{(h)}, \tilde{\mathbf{v}}^{(h)} \right\rangle v^{(h)} \tag{31}$$

$$\overset{3}{=} N^4 \|\tilde{\mathbf{v}}\|^4 \sum_{h=1}^{H} \left\langle \tilde{\mathbf{v}}^{(h)}, \tilde{\mathbf{v}}^{(h)} \right\rangle v^{(h)}, \tag{32}$$

where we define $\tilde{\mathbf{v}} = \sum_{h=1}^{H} \tilde{\mathbf{v}}^{(h)}$. Equality (1) is because in both $W^{K,3,h}$ and $W^{Q,3,h}$ on the first row is nonzero, and it has ones everywhere except in coordinates that are multiples of $d_a$, resulting in summing over all of these non-zero elements of the vector $\mathbf{y}^{(2)}$. Equality (2) is because in the vector $\mathbf{v}^{(h)}$ every entry has a corresponding entry equal to its complement, which upon summation is equal to one, leaving only the $\left\langle \tilde{\mathbf{v}}^{(h')}, \tilde{\mathbf{v}}^{(h')} \right\rangle$ coefficients of the vector $\mathbf{y}^{(2)}$. Equality (3) is because

$$\|\tilde{\mathbf{v}}\|^2 = \langle \tilde{\mathbf{v}}, \tilde{\mathbf{v}} \rangle = \sum_{h_1,h_2} \left\langle \tilde{\mathbf{v}}^{(h_1)}, \tilde{\mathbf{v}}^{(h_2)} \right\rangle = \sum_{h=1}^{H} \left\langle \tilde{\mathbf{v}}^{(h)}, \tilde{\mathbf{v}}^{(h)} \right\rangle, \tag{33}$$

where the last equality stems from the fact that every $\tilde{\mathbf{v}}^{(h)}$ is non-zero on a different segment of its $d_x$ coordinates.

For any subsequent layer $l < L$ we use the same set of parameters, and since the input of each preceding layer has the same form of $\mathbf{y}^{(l)} = N^{\alpha_l} \cdot \|\tilde{\mathbf{v}}\|^{2\beta_l} \sum_{h=1}^{H} \left\langle \tilde{\mathbf{v}}^{(h)}, \tilde{\mathbf{v}}^{(h)} \right\rangle \mathbf{v}^{(h)}$, then we can just compute its recurrence relation:

$$\mathbf{y}^{(l+1)} = N \sum_{h=1}^{H} \left( N^{\alpha_l} \|\tilde{\mathbf{v}}\|^{2\beta_l} \sum_{h'=1}^{H} \left\langle \tilde{\mathbf{v}}^{(h')}, \tilde{\mathbf{v}}^{(h')} \right\rangle \right)^2 N^{\alpha_l} \|\tilde{\mathbf{v}}\|^{2\beta_l} \left\langle \tilde{\mathbf{v}}^{(h)}, \tilde{\mathbf{v}}^{(h)} \right\rangle v^{(h)} \tag{34}$$

$$= N^{1+3\alpha_l} \|\tilde{\mathbf{v}}\|^{6\beta_l} \sum_{h=1}^{H} \left( \sum_{h'=1}^{H} \left\langle \tilde{\mathbf{v}}^{(h')}, \tilde{\mathbf{v}}^{(h')} \right\rangle \right)^2 \left\langle \tilde{\mathbf{v}}^{(h)}, \tilde{\mathbf{v}}^{(h)} \right\rangle v^{(h)} \tag{35}$$

$$= N^{3\alpha_l+1} \|\tilde{\mathbf{v}}\|^{2\cdot(3\beta_l+2)} \sum_{h=1}^{H} \left\langle \tilde{\mathbf{v}}^{(h)}, \tilde{\mathbf{v}}^{(h)} \right\rangle v^{(h)} \tag{36}$$

$$\Rightarrow \alpha_{l+1} = 3\alpha_l + 1, \beta_{l+1} = 3\beta_l + 2 \tag{37}$$

Using the initial conditions of $\alpha_3 = 4$ and $\beta_3 = 2$, we get that $\alpha_l = \frac{3^{l-1}-1}{2}, \beta_l = 3^{l-2} - 1$. For the $L$'th layer, the only difference is that $W^{V,L,h}$ is defined such that it returns a 1-hot vector that picks the $d_a$'th element of the previous step. Putting it all together we get:

$$y_k^{(L)} = N^{\frac{3^{L-1}-1}{2}} \cdot \|\tilde{\mathbf{v}}\|^{2\cdot(3^{l-2}-1)} \sum_{h=1}^{H} \left\langle \tilde{\mathbf{v}}^{(h)}, \tilde{\mathbf{v}}^{(h)} \right\rangle \mathbf{i} \cdot v_{d_a}^{(h)} \tag{38}$$

$$y_k^{(L)} = N^{\frac{3^{L-1}-1}{2}} \cdot \mathbf{i} \cdot \|\tilde{\mathbf{v}}\|^{2\cdot3^{l-2}} \tag{39}$$

Finally, we can evaluate $\|\tilde{\mathbf{v}}\|^2$:

$$\|\tilde{\mathbf{v}}\|^2 = \sum_{k=1}^{d_x} \tilde{v}_k^2 = \sum_{h=1}^{H} \sum_{k=1}^{d_a-1/2} \left(V_{i_1,(d_a-1)\cdot(h-1)+k} + \mathbf{i} \cdot V_{j_1,(d_a-1)\cdot(h-1)+k}\right)^2 \tag{40}$$

$$= \overbrace{\sum_{h=1}^{H} \sum_{k=1}^{d_a-1/2} V_{i_1,(d_a-1)\cdot(h-1)+k}^2}^{\text{normalized}\Rightarrow=1} - \overbrace{\sum_{h=1}^{H} \sum_{k=1}^{d_a-1/2} V_{j_1,(d_a-1)\cdot(h-1)+k}^2}^{\text{normalized}\Rightarrow=1} \tag{41}$$

$$2\mathbf{i} \cdot \sum_{h=1}^{H} \sum_{k=1}^{d_a-1/2} V_{i_1,(d_a-1)\cdot(h-1)+k} V_{j_1,(d_a-1)\cdot(h-1)+k} \tag{42}$$

$$= 2\mathbf{i}(VV^T)_{i_1,j_1}, \tag{43}$$

which concludes the proof. $\qquad\square$

Next, we show two lemmas that aid in the proof of the lower bound. We first quote an identity by which for a matrix with entries that are polynomials in $x$, if a single contributor to the determinant has the highest degree of $x$, then the matrix is fully ranked for all values of $x$ but a finite set.

**Lemma 7.** *(from Levine et al. [2018]). Let $A \in \mathbb{R}^{N \times N}$ be a matrix whose entries are polynomials in $x \in \mathbb{R}$. In this case, its determinant may be written as $\det(A) = \sum_{\sigma \in S_N} sgn(\sigma)p_\sigma(x)$, where $S_N$ is the symmetric group on $N$ elements and $p_\sigma(x)$ are polynomials defined by $p_\sigma(x) \equiv \prod_{i=1}^{N} A_{i\sigma(i)}(x), \forall \sigma \in S_n$. Additionally, let there exist $\bar{\sigma}$ such that $\deg(p_{\bar{\sigma}}(x)) > \deg(p_\sigma(x)) \forall \sigma \neq \bar{\sigma}$. Then, for all values of $x$ but a finite set, $A$ is fully ranked.*

*Proof.* We show that in this case $\det(A)$, which is a polynomial in $x$ by its definition, is not the zero polynomial. Accordingly, $\det(A) \neq 0$ for all values of $x$ but a finite set. Denoting $t \equiv \deg(p_{\bar{\sigma}}(x))$, since $t > \deg(p_\sigma(x)) \forall \sigma \neq \bar{\sigma}$, a monomial of the form $c \cdot x^t, c \in \mathbb{R} \setminus \{0\}$ exists in $p_{\bar{\sigma}}(x)$ and doesn't exist in any $p_\sigma(x)$, $\sigma \neq \bar{\sigma}$. This implies that $\det(A)$ is not the zero polynomial, since its leading term has a non-vanishing coefficient $sgn(\bar{\sigma}) \cdot c \neq 0$, and the lemma follows from the basic identity: $\det(A) \neq 0 \iff A$ is fully ranked. $\qquad\square$

The following quoted lemma, establishes a relation referred to as the *vector rearrangement inequality*, which helped us ensure that our matrix of interest upholds the conditions of lemma 7 and is thus fully ranked.

**Lemma 8.** *(from Levine et al. [2018]). Let $\{\mathbf{v}^{(i)}\}_{i=1}^{N}$ be a set of $N$ different vectors in $\mathbb{R}^{\bar{R}}$ such that $\forall i \in [N], j \in [\bar{R}] : v_j^{(i)} \geq 0$. Then, for all $\sigma \in S_N$ such that $\sigma \neq \mathbb{I}_N$, where $S_N$ is the symmetric group on $N$, it holds that:*

$$\sum_{i=1}^{N} \left\langle \mathbf{v}^{(i)}, \mathbf{v}^{(\sigma(i))} \right\rangle < \sum_{i=1}^{N} \left\| \mathbf{v}^{(i)} \right\|^2.$$

*Proof.* We rely on theorem 368 in [Hardy et al., 1952], which implies that for a set of non-negative numbers $\{a^{(1)}, \ldots, a^{(N)}\}$ the following holds for all $\sigma \in S_N$:

$$\sum_{i=1}^{N} a^{(i)} a^{(\sigma(i))} \leq \sum_{i=1}^{N} (a^{(i)})^2, \tag{44}$$

with equality obtained only for $\sigma$ which upholds $\sigma(i) = j \iff a^{(i)} = a^{(j)}$. The above relation, referred to as the *rearrangement inequality*, holds separately for each component $j \in [\bar{R}]$ of the given vectors:

$$\sum_{i=1}^{N} v_j^{(i)} v_j^{(\sigma(i))} \leq \sum_{i=1}^{N} (v_j^{(i)})^2.$$

We now prove that for all $\sigma \in S_N$ such that $\sigma \neq \mathbb{I}_N$, $\exists \hat{j} \in [\bar{R}]$ for which the above inequality is hard, *i.e.*:

$$\sum_{i=1}^{N} v_{\hat{j}}^{(i)} v_{\hat{j}}^{(\sigma(i))} < \sum_{i=1}^{N} (v_{\hat{j}}^{(i)})^2. \qquad (45)$$

By contradiction, assume that $\exists \hat{\sigma} \neq \mathbb{I}_N$ for which $\forall j \in [\bar{R}]$:

$$\sum_{i=1}^{N} v_{j}^{(i)} v_{j}^{(\hat{\sigma}(i))} = \sum_{i=1}^{N} (v_{j}^{(i)})^2.$$

From the conditions of achieving equality in the rearrangement inequality defined in Equation (44), it holds that $\forall j \in [\bar{R}] : v_{j}^{(\hat{\sigma}(i))} = v_{j}^{(i)}$, trivially entailing: $\mathbf{v}^{(\hat{\sigma}(i))} = \mathbf{v}^{(i)}$. Thus, $\hat{\sigma} \neq \mathbb{I}_N$ would yield a contradiction to $\{\mathbf{v}^{(i)}\}_{i=1}^{N}$ being a set of $N$ different vectors in $\mathbb{R}^{\bar{R}}$. Finally, the hard inequality of the lemma for $\sigma \neq \mathbb{I}_N$ is implied from Equation (45):

$$\sum_{i=1}^{N} \left\langle \mathbf{v}^{(i)}, \mathbf{v}^{(\sigma(i))} \right\rangle \equiv \sum_{i=1}^{N} \left( \sum_{j=1}^{\bar{R}} v_{j}^{(i)} v_{j}^{(\sigma(i))} \right) = \sum_{j=1}^{\bar{R}} \left( \sum_{i=1}^{N} v_{j}^{(i)} v_{j}^{(\sigma(i))} \right) < \sum_{j=1}^{\bar{R}} \left( \sum_{i=1}^{N} (v_{j}^{(i)})^2 \right) = \sum_{i=1}^{N} \left\| \mathbf{v}^{(i)} \right\|^2 .$$

$\square$

## C    Proof of Proposition 1 on the separation rank symmetry

**Claim 3.** *For any depth $L \geq 1$ input size $N > 1$ and output locations $i \in [N]$, $p \in [d_x]$ The separation rank w.r.t. balanced partitions, which obey $A \cup B = [N], |A|, |B| = {}^N/_2$, is invariant to the identity of the partition, i.e., $\forall A \cup B = [N], \tilde{A} \cup \tilde{B} = [N], \; s.t. \; |A|, |B|, |\tilde{A}|, |\tilde{B}| = {}^N/_2$:*

$$sep(y_p^{i,L,d_x,H,\Theta}; A, B) = sep(y_p^{i,L,d_x,H,\Theta}; \tilde{A}, \tilde{B}) \qquad (46)$$

*Proof.* We will denote $A = \left( a_1, \ldots, a_{\frac{N}{2}} \right), B = \left( b_1, \ldots, b_{\frac{N}{2}} \right), \tilde{A} = \left( \tilde{a}_1, \ldots, \tilde{a}_{\frac{N}{2}} \right), \tilde{B} = \left( \tilde{b}_1, \ldots, \tilde{b}_{\frac{N}{2}} \right)$ and by $\pi \in S_N$ the unique permutation that satisfy

$$\forall m \in \left[ \frac{N}{2} \right] \quad \pi(a_m) = \tilde{a}_m \wedge \pi(b_m) = \tilde{b}_m$$

w.l.o.g we will assume that $a_1 = \tilde{a}_1 = i$.

Assuming that $sep(y; A, B) = R$, then there exist $g_1, \ldots, g_R, g_1', \ldots, g_R'$ s.t.

$$\forall \mathbf{x}^{(1)}, \ldots, \mathbf{x}^{(N)} \in \mathbb{R}^{d_x} \quad y_p^{i,L,d_x,H,\Theta} \left( \mathbf{x}^{(1)}, \ldots, \mathbf{x}^{(N)} \right) = \sum_{v=1}^{R} g_v \left( \mathbf{x}^{(a_1)}, \ldots, \mathbf{x}^{\left( a_{\frac{N}{2}} \right)} \right) g_v' \left( \mathbf{x}^{(b_1)}, \ldots, \mathbf{x}^{\left( b_{\frac{N}{2}} \right)} \right)$$

$i = \pi(a_1) = a_1$ therefore the summations over $j_1, \ldots, j_N$ in eq. (5) implies that for any $x^{(1)}, \ldots, x^{(N)} \in \mathbb{R}^{d_x}$ we have

$$y_p^{i,L,d_x,H,\Theta} \left( \mathbf{x}^{(1)}, \ldots, \mathbf{x}^{(N)} \right) = y_p^{i,L,d_x,H,\Theta} \left( \mathbf{x}^{(\pi(1))}, \ldots, \mathbf{x}^{(\pi(N))} \right)$$

And therefore

$$= \sum_{v=1}^{R} g_v \left( \mathbf{x}^{(\pi(a_1))}, \ldots, \mathbf{x}^{\left( \pi\left( a_{\frac{N}{2}} \right) \right)} \right) g_v' \left( \mathbf{x}^{(\pi(b_1))}, \ldots, \mathbf{x}^{\left( \pi\left( b_{\frac{N}{2}} \right) \right)} \right)$$

$$= \sum_{v=1}^{R} g_v \left( \mathbf{x}^{(\tilde{a}_1)}, \ldots, \mathbf{x}^{\left( \tilde{a}_{\frac{N}{2}} \right)} \right) g_v' \left( \mathbf{x}^{(\tilde{b}_1)}, \ldots, \mathbf{x}^{\left( \tilde{b}_{\frac{N}{2}} \right)} \right)$$

So we proved that

$$sep(y_p^{i,L,d_x,H,\Theta}; \tilde{A}, \tilde{B}) \leq sep(y_p^{i,L,d_x,H,\Theta}; A, B)$$

Finally by switching the roles of $\tilde{A}, \tilde{B}$ and $A, B$ we can get the inverse inequality so we conclude that

$$sep(y_p^{i,L,d_x,H,\Theta}; \tilde{A}, \tilde{B}) = sep(y_p^{i,L,d_x,H,\Theta}; A, B)$$

$\square$

# D   Experimental details

We trained common self-attention architectures of depths $L = 6, 12, 18, 24, 30, 36, 48$ and varying widths, such that the network sizes range between $1.1 \cdot 10^6$ and $5.8 \cdot 10^8$ (full details on the widths of the trained architectures are given in the appendix). We trained decoder-only (unidirectional) models, by optimizing the autoregressive log-likelihood of the training examples. We used a smaller than usual vocabulary size of 2000 so that the vocabulary embedding parameters, given by $d_x \cdot V$ for a vocabulary of size $V$, would constitute a small fraction of the learned parameters for all data points. Autoregressive models were shown to work well even on character level vocabularies (*e.g.*, [Peters et al., 2018]); due to modeling a joint distribution over the text, they are less sensitive to vocabulary size than bidirectional models [Levine et al., 2020].

Our training set was English Wikipedia, BookCorpus and OpenWebText. We report the loss on a held out test set of size 170K sequences. Notably, we estimated the variance of the pretraining and evaluation procedure by rerunning 11 of the trained architectures three times each, and found it to be very low – the reported test loss is stable up to its third digit. The remainder of the training details are given in the appendix.

We conducted the training with Adam optimizer for $1M$ steps and a batch size of 512 sequences of 128 tokens. All experiments used a learning rate schedule with a 12000 step linear warm-up followed by a cosine decay to zero. In order to increase width without changing other architectural parameters, we kept the number of heads per layer constant at 2 (experimental evidence indicates that many heads per layer are not crucial [Michel et al., 2019, Kaplan et al., 2020], as does our theoretical analysis which shows that the number of heads per layer affects the separation rank logarithmically).

Table 1 shows the per-depth widths of the trained architecture. More experiments were conducted per adjacent depth pairs in order to identify the transition point accurately, and reduce the error bars in figure 3a of the main text. Table 2 details the different standard deviation of repeating the training and evaluation experiment 3 times per the given architectures.

Arash Amini, Amin Karbasi, and Farokh Marvasti. Low-rank matrix approximation using point-wise operators. *IEEE Transactions on Information Theory*, 58(1):302–310, 2012.

Richard Caron and Tim Traynor. The zero set of a polynomial. *WSMR Report 05-02*, 2005.

Wolfgang Hackbusch. *Tensor spaces and numerical tensor calculus*, volume 42. Springer Science & Business Media, 2012.

Godfrey Harold Hardy, John Edensor Littlewood, and George Pólya. *Inequalities*. Cambridge university press, 1952.

Jared Kaplan, Sam McCandlish, Tom Henighan, Tom B Brown, Benjamin Chess, Rewon Child, Scott Gray, Alec Radford, Jeffrey Wu, and Dario Amodei. Scaling laws for neural language models. *arXiv preprint arXiv:2001.08361*, 2020.

Yoav Levine, Or Sharir, Alon Ziv, and Amnon Shashua. Benefits of depth for long-term memory of recurrent networks. *(ICLR 2018) International Conference on Learning Representations workshop*, 2018.

Yoav Levine, Barak Lenz, Opher Lieber, Omri Abend, Kevin Leyton-Brown, Moshe Tennenholtz, and Yoav Shoham. Pmi-masking: Principled masking of correlated spans. *arXiv preprint arXiv:2010.01825*, 2020.

Paul Michel, Omer Levy, and Graham Neubig. Are sixteen heads really better than one? In *Advances in Neural Information Processing Systems*, pages 14014–14024, 2019.

Matthew E Peters, Mark Neumann, Mohit Iyyer, Matt Gardner, Christopher Clark, Kenton Lee, and Luke Zettlemoyer. Deep contextualized word representations. *arXiv preprint arXiv:1802.05365*, 2018.

Or Sharir, Ronen Tamari, Nadav Cohen, and Amnon Shashua. Tractable generative convolutional arithmetic circuits. 2016.

Andreas Veit, Michael J Wilber, and Serge Belongie. Residual networks behave like ensembles of relatively shallow networks. In *Advances in neural information processing systems*, pages 550–558, 2016.

| L=6 | L=12 | L=18 | L=24 | L=30 | L=36 | L=48 |
|---|---|---|---|---|---|---|
| 128 | 88 | - | 64 | - | - | 44 |
| 168 | 120 | - | 88 | - | - | 60 |
| 216 | 152 | - | 104 | - | - | 72 |
| 220 | 156 | - | - | - | - | - |
| 224 | 160 | - | 112 | - | - | 80 |
| 248 | 176 | - | 128 | - | - | 88 |
| 272 | 192 | - | 136 | - | - | 96 |
| 296 | 208 | - | 144 | - | - | 104 |
| 320 | 224 | 184 | 160 | 144 | 128 | 112 |
| 376 | 264 | 216 | 184 | 168 | 152 | 128 |
| - | 272 | 244 | - | - | - | - |
| 408 | 288 | 232 | 200 | 176 | 160 | 144 |
| - | 296 | 240 | - | - | - | - |
| - | 304 | 248 | - | - | - | - |
| - | 314 | 256 | - | - | - | - |
| 456 | 320 | 264 | 224 | 200 | 184 | 160 |
| 496 | 352 | 288 | 248 | 224 | 200 | 176 |
| 568 | 400 | 320 | 280 | 248 | 232 | 200 |
| 680 | 480 | 384 | 336 | 304 | 272 | 240 |
| - | - | 398 | 344 | - | - | - |
| - | - | 406 | 352 | - | - | - |
| - | - | 416 | 360 | - | - | - |
| - | - | 424 | 368 | - | - | - |
| - | - | 440 | 376 | - | - | - |
| 816 | 576 | 472 | 408 | 368 | 336 | 288 |
| 960 | 680 | 560 | 480 | 432 | 392 | 336 |
| 1088 | 768 | 624 | 544 | 484 | 440 | 384 |
| - | - | - | 584 | 520 | - | - |
| 1416 | 1000 | - | 704 | 632 | 576 | 496 |
| - | - | - | - | 808 | 736 | - |
| 2128 | 1504 | - | 1064 | 952 | 872 | 752 |
| 2832 | 2000 | - | 1416 | - | 1160 | 1000 |

Table 1: The widths $d_x$ of the different trained networks.

| $d_x = 320$ | $d_x = 680$ | $d_x = 800$ |
|---|---|---|
| 1.92E-03 | 2.06E-03 | 6.51E-04 |

(a) L = 6

| $d_x = 224$ | $d_x = 400$ | $d_x = 680$ | $d_x = 1000$ |
|---|---|---|---|
| 2.08E-03 | 1.65E-03 | 1.33E-03 | 1.20E-03 |

(b) L = 12

| $d_x = 160$ | $d_x = 280$ | $d_x = 480$ | $d_x = 704$ |
|---|---|---|---|
| 7.36E-04 | 1.02E-03 | 1.48E-03 | 7.76E-04 |

(c) L = 24

Table 2: The standard deviation of the test loss for networks of varying widths and depths, when repeating the training and evaluation experiment 3 times per point.

## Footnotes

[1]We have embedded the Feed-Forward layer within $W^O$ due to the linearity of the analyzed model.