[Reviews · NeurIPS 2020]

Review 1

Summary and Contributions: (post-rebuttal) I have read the authors' response. Please do make sure to include corrections/updates that are mentioned in the rebuttal, especially the residual connections. My belief is that this paper makes interesting and nontrivial contributions on the width-depth tradeoff of Transformers; thus, I have raised the score. --------------------------------- This paper studies the expressive power of self-attention models, which are very popular in natural language processing. The paper focuses on a simplified model where the ReLU activations, softmax operation, and layer normalization are removed from the standard BERT architecture. Using the concept of the “separation rank” which measures how “difficult” it is to capture the dependency between two sets of input variables, the paper shows that the growth of the separation rank as a function of the embedding dimension (or “width”) dx and depth L exhibits a phase transition: 1. If L < log_3 dx, the separation rank grows doubly exponentially in depth L but polynomially in width dx. 2. If L > log_3 dx, the separation rank has an upper bound that is exponential in both depth L and width dx. These results show that, up to a certain depth threshold logarithmic in the width, the depth is much more efficient than the width in improving the expressive power of self-attention models. However, after the threshold, the depth-efficiency becomes limited and the width and the depth contribute similarly to the expressive power. While this paper only presents theoretical results, the theory is supported by empirical observations done in a different paper.

Strengths: This paper reveals an interesting phenomenon that for self-attention models, the efficiency of depth is rather limited past a certain threshold. The theory of self-attention models is lagged behind compared to its huge empirical success in NLP, so the concrete theoretical results presented in this paper are timely and much-needed. I lowered my score for now because of some clarifications I need, but I’m happy to raise my score depending on the authors’ response.

Weaknesses: The weakness of this paper lies in the fact that they study a simplified model without any ReLU activation function and softmax operators. However, given that linear neural networks are also a popular subject of study in the theory literature, I don’t consider this a big weakness.

Correctness: As far as I can tell from the main text, the claims look correct, except for some clarification issues that I will list below. I also briefly checked the proof of the upper bounds in the supplementary material.

Clarity: The presentation of the paper is good overall, but there are certain parts that have room for improvement, in my opinion. In particular, I believe that the paper would benefit from moving some of the details deferred to the supplementary material back to the main text. For example: - The separation rank is the main subject of study in this paper, so formal definition should better appear in the main text. - In Theorems 1 and 2, the dependency of constants a_i, b_i, c_i on L and log (dx) should better be explicitly spelled out, because L and log (dx) are indeed the key variables of the bound. - Equation (4) interrupted my flow of reading this paper: why specifically C = (3^L - 1)/2, what is g^L, why are there so many arguments in g^L, etc.? A more precise description of g^L rather than a “placeholder” would be helpful. In fact, Eq (4) is rarely mentioned in the main text; maybe it can be deferred to the supplementary material or to the proof sketch?

Relation to Prior Work: To the best of my knowledge, this paper does a good job summarizing and citing existing results.

Reproducibility: Yes

Additional Feedback: I have some clarification questions: - In Line 115, the paper claims that the feed-forward layer and the residual connections can be embedded within W^{O,l,h}. While I agree that the feed-forward layer is just a multiplication of a matrix when ReLU is taken out, I do not fully understand why this is the case for residual connections. The i-th position of output of the self-attention layer (including the skip connection) reads x^{l,i}+f^{l,i}_{SA}, and after removing the softmax, this will be of the form x^{l,i}+(a degree 3 polynomial of inputs). In other words, using the notation X \in R^{dx \times N} as in the supplementary material (assuming H=1 for simplicity), the self-attention layer and the residual connection is X + W^O W^V X X^T (W^K)^T W^Q X. Given that the second term has three X’s, I do not see how one can embed the skip connection to W^O. - It looks like the upper bounds (Theorems 1.1 and 1.2 in the supplementary material) do not depend on whether L < log_3 dx or L > log_3 dx. Denoting the upper bounds in Theorems 1.1 and 1.2 by b1(L) and b2(L), respectively, my understanding is that the “phase transition” occurs because min(b1(L), b2(L)) = b1(L) for L < log_3 dx and min(b1(L), b2(L)) = b2(L) for L > log_3 dx. However, after examining the exact coefficients presented in the proofs for L = log_3 dx, it seems that the 2dx log_3 dx terms cancel out in b2(log_3 dx), so b1(log_3 dx) = O(dx log dx) but b2(log_3 dx) = O(dx + log dx). This would mean that the threshold is in fact smaller than L = log_3 dx. Is there any particular reason as to why the authors chose to claim L = log_3 dx as the threshold? Is it because the lower bound in Theorem 1 holds for L < log_3 dx only? - Corollary 1 says that d_x^{shallow} \propto \exp(\exp(L^{deep})) is necessary to represent a function realized by a deep model with a shallow model satisfying L^{shallow} = \alpha L^{deep} for some small \alpha. I wonder why there is no dependence on \alpha in the requirement for d_x^{shallow} \propto \exp(\exp(L^{deep})). Is the width requirement the same for very small \alpha and relatively big \alpha? Additional questions, disregard if you don’t have enough space for rebuttal: - Can you show a matching lower bound in Theorem 2? - Theorem 1 requires that H > 1, i.e., the self-attention layers have multiple heads. Does it mean that the theory does not hold for single-head attention models? How is the growth of separation rank like in case of single-head models?


Review 2

Summary and Contributions: UPDATE: Thanks to the author(s) for the response. I still believe that some well-designed toy experiments can be illuminating, and can improve the paper very much. Personally, I was a bit disappointed to try a wide BERT and have it not perform well at all, after reading your paper. I think it'd be wise to improve your paper so other readers won't feel the same way as well. The paper explores the benefit of width vs depth in a transformer through the lens of expressivity. They show that the separation rank of a transformer increases fast with depth when depth is below log(width); otherwise, depth and width contribute similarly to separation rank. They also support their argument by citing a figure from Kaplan et al. 2020. The paper suggests transformers should have depth L be equal to log3(width).

Strengths: As people are training larger and larger transformers, hyperparameter tuning becomes more costly, and any theoretical guidance on hyperparameter choices are desired. This paper tries to tackle the issue of width vs depth allocation in a transformer. This is thus a timely paper and potentially can have a lot of impact. The proof insights are, to my knowledge, novel.

Weaknesses: 1. My major confusion is the explanation of Kaplan et al.’s figure: Shouldn’t you want to say, for any fixed vertical slice (fixing #params), the performance saturates at a depth that’s predicted by your equation? This should be explained a lot more clearly, in the figure caption and in the text. 2. Another concern is Theorem 1 applies to “almost every” weight assignment. However, it’s highly likely that after training, the network will converge to a low dimensional submanifold of the weight space (c.f. Ji & Telgarsky). Therefore, I don’t know how much I should trust this result is applicable to trained networks. More minor questions 3. According to the recommendation of your table 1, I reshaped BERT-large to be shallower but wider, with the same number of parameters. However, the performance is much worse than the original model. This suggests that perhaps the evidence of Fig. 1 is more circumstantial and dependent on other hyperparameters, and optimization is still a key factor to performance, which is not discussed in this paper. Can you say anything about optimization? 4. Is there a way to measure the separation rank of real networks? It would be great to see some experiments 5. What’s the role of heads? It’s always polynomial in width? 6. Section 2.1: “BERT” should be replaced with “Transformer” 7. Line 120-122: sentence long and confusing 8. Why is claim 1 not a theorem or proposition? “Claim” suggests that it’s a heuristic argument. Is that true? 9. What’s the dependence on N in Theorem 1? 10. Line 242-244: explain the rank argument more carefully. Ji, Ziwei, and Matus Telgarsky. "Gradient descent aligns the layers of deep linear networks." arXiv preprint arXiv:1810.02032 (2018).

Correctness: All of the theoretical arguments seem correct. However, because there are some claims about the implication of this in practice, I’d like to see some carefully designed experiments beyond the figure from Kaplan et al.

Clarity: The overall picture is relatively clear, but as I explained above, there are still some areas to improve on clarity.

Relation to Prior Work: This work cites and builds on prior work well, to my knowledge.

Reproducibility: Yes

Additional Feedback:


Review 3

Summary and Contributions: -------------------------- UPDATE: After reading the authors's response, I still believe that the paper could be improved by doing some experiments on the Transformer models or on the linearized self-attention models to verify the theoretical analysis. R2's result on BERT might also show that the theoretical analysis on the linearized self-attention model might not agree with empirical findings on the real Transformer model. But I do think it's an interesting paper and the theoretical results could shed some light on the width-depth tradeoff for Transformer models. Thus, I keep my score as 5 but I'm also feeling good if the paper is finally accepted. -------------------------- This paper focuses on the interplay between depth and width for Transformer models. It studies a simplified model where all non-linear activations and normalizations are removed to analyze the bottleneck of stacking self-attention layers in terms of modeling input dependencies (as measured by separation rank). It theoretically establishes an interesting result that there exists a depth threshold for self-attention layers which depends logarithmically on the width and increasing depth is more efficient than width below this threshold. The theoretical result is shown to be consistent with the experimental result in Kaplan et al., 2020 and has practical implications for self-attention model design.

Strengths: + The problem is very motivated since the depth efficiency for Transformer models is not clearly observed in practice, in contrast to other deep learning models. + This paper establishes some interesting theoretical results about the limitation of depth efficiency for self-attention models, from the perspective of the function’s separation rank bottleneck by stacking self-attention layers. + The theoretical result has practical implications for parameter allocation between depth and width for self-attention models.

Weaknesses: - The problem setting of this paper is too simplified, where only a “linearized” self-attention layer with all non-linear activations, layer normalization and softmax operation removed. However, given that the main purpose of the paper is to analyze the functionality of self-attention in terms of integrating inputs, these relaxations are not totally unreasonable. - The experiments are not sufficient. More empirical experiments or toy experiments (for the simplified self-attention model considered in the theoretical analysis) need to be done to show the validity of the model relaxations and the consistence of the theoretical analysis with empirical results, besides citing the result in Kaplan et al. 2020. - Although the paper is well organized, some parts are not well explained, especially for the proof sketch for Theorem 1 and Theorem 2.

Correctness: I don’t find out an obvious methodological mistake in the paper.

Clarity: Overall, the paper is well organized. But some parts are not well explained, especially for the proof sketch for Theorem 1 and Theorem 2.

Relation to Prior Work: This paper doesn’t discuss about the related work very comprehensively. Actually, it is not clear that the depth inefficiency of Transformer models results from the expressivity of stacking self-attention layers (as discussed in this paper) or the difficulty of training deep Transformer models (e.g., Huang et al. [1]). It would be great if the paper discusses more about these related literatures. [1] http://www.cs.toronto.edu/~mvolkovs/ICML2020_tfixup.pdf

Reproducibility: Yes

Additional Feedback: - Since in Tab. 1 shows the depth threshold for different model size, it would be great to carry out more experiments for larger model size (10^9 ~ 10^11) and more fine-grained layers (e.g., 6~12 layers), and plot the theoretical depth threshold and empirical result as in Fig. 1 to see if they are consistent. - Although the separation rank may be a good theoretical metric for measuring the ability to model input dependencies, are there any empirical evidences that it could indeed predict the model expressivity of the self-attention model? - In Eq. 2, there should be a layer normalization after the feedforward sub-layer as well.


Review 4

Summary and Contributions: This paper aims at providing fundamental theory to address the question of the depth to width trade-off in self-attention networks. Some findings are interesting and maybe valuable for future research.

Strengths: The motivation of the paper is clear: provides the fundamental theory to understand the trade-off between depth and width in the self-attention networks. The theory and proof look sound and reasonable.

Weaknesses: The study is conducted on the self-attention networks in which all non-linear activations and normalization operations are removed. It seems not the reflection of the real self-attention models. More analysis of those removals should be done.

Correctness: The claims in the paper is technically correct and in line with many empirical studies.

Clarity: The paper is well written with sufficient proofs.

Relation to Prior Work: It is clearly discussed how this work differs from previous contributions.

Reproducibility: Yes

Additional Feedback:

[Author Response · NeurIPS 2020]

We thank all reviewers for their thoughtful feedback, which aided us in sharpening the presentation of our results.
Following **R1**'s questions on bounds, we will present them more explicitly in the paper, as briefly described here.
Coefficients in Th1: Combining the lower bound stated in Th2.1 in the Supplementary Material (SM), with the upper
bound in line 55 in the SM, Th1 will explicitly state: $3^{L-2}\left(\log_3\left(d_x - H\right) + a\right) \leq \log_3 sep(y) \leq \frac{3^L - 1}{2}\log_3\left(d_x + H\right)$
with $a = -L + [2 - \log_3 2]$. Corollary 1: As we note in lines 163-167 of the SM, the above lower bound is
tight w.r.t. the dependence on $\overline{\text{depth and}}$ width, meets and improves upon the dependence stated in the lower
bound of Th1 in the main text, and consequently improves also on the corollary. We refer R1 to corollary 2.1
in lines 179-183 of the SM, which we will place instead of corollary 1, and which fully addresses their question.
Regime transition point (and lower bound in Th2): In lines 205-208 of the SM we show that the separation rank is
lower bounded by $\left(\!\!\binom{(d_x - H)/2}{3^{L-2}}\!\!\right)$. By using the dual identity: $\left(\!\!\binom{n}{k}\!\!\right) = \binom{n+k-1}{k} = \binom{n+k-1}{n-1}$, and also $\binom{a}{b} \geq \left(\frac{a}{b}\right)^b$, we
get: $\left(\!\!\binom{(d_x - H)/2}{3^{L-2}}\!\!\right) \geq \max\{\left(\frac{(d_x - H)/2 - 1}{3^{L-2}} + 1\right)^{3^{L-2}}, \left(\frac{3^{L-2}}{(d_x - H)/2 - 1} + 1\right)^{(d_x - H)/2 - 1}\}$. From the symmetry of $n - 1$ and
$k$ in the definition of $\left(\!\!\binom{n}{k}\!\!\right)$, the transition of lower bounds occurs at $d_x/2 \simeq 3^{L-2} \to L \simeq \log_3 d_x + 1.3$ (neglecting
$H << d_x$). Regarding upper bounds, R1's question aided us in finding a typo in eq. 6 of the SM (remnant from an ear-
lier version): in the transition from the second to the third line of eq. 6, a plus 1 was mistakenly omitted. Recalling that
$C(L) = \frac{3^L - 1}{2}$, the correct continuation is that the second line in eq. 6 of the SM $\leq \log_3[3^L d_x (2e)^{d_x} (\frac{3^L - 1}{d_x} + \mathbf{1})^{2d_x}]$.
From here, *only for* $3^L > d_x$, the upper bound is $\log_3 sep(y) \leq \log_3[3^L d_x (2e)^{d_x} (2 \cdot \frac{3^L - 1}{d_x})^{2d_x}]$. Coefficients in Th2:
Combining this upper bound with the lower bound above (right term in the max), Th2 is also tight w.r.t. lead-
ing terms of depth and width, and will explicitly state: $\frac{1}{2}d_x \cdot L + b_1 + b_2 \leq \log_3 sep(y) \leq 2d_x \cdot L + c_1 + c_2$
with corrections on the order of $L$: $b_1 = -L\left(\frac{H}{2} + 1\right), c_1 = L$ and corrections on the order of $d_x \log_3(d_x)$:
$b_2 = -d_x\left(1 + \frac{1}{2}\log_3\left(\frac{d_x - H}{2}\right)\right) c_2 = -2d_x \cdot \log_3 d_x/2\sqrt{2e} + \log_3 d_x$. Residual connections: we thank R1 for cor-
rectly pointing out that the residual connection cannot be embedded in the output matrix. Since it was not part of the
core attention operation we neglected it too hastily. This functionality is easily embedded in our approach - an upper
bound on the separation rank of a depth $L$ network with residual connections is $2^L$ times the proven separation ranks
without it, which means adding a factor of $L \log_3 2$ to the upper bounds on $\log_3 sep(y)$ above. Upper bound is the
relevant concern here (ensuring that the skip connections don't boost expressiveness), and the lower bound, which
almost covers this case in its current form, will be similarly minorly tweaked to include this functionality.

Following **R2** and **R3**'s questions: our contribution focuses solely on expressiveness aspects which draw the boundaries
of what is achievable for any optimization. Indeed, having proven the results for all configurations but a set of measure
zero theoretically leaves a chance for all configurations of interest to reside within that measure zero subspace. While
we agree with R2 that trained networks are likely to reside in a low dimensional submanifold of parameter space, it is
not clear that these two will contain each other or even intersect (the measure zero in our derivation is due to zeros of
a polynomial dictated by the architecture, not related to any specific type of data). In fact, we view the experimental
evidence in fig.1 as contradicting the possibility that the measure zero exception occurs in relevant functions – trained
networks seem to exhibit the behaivior depicted by our "almost everywhere" trends. Note that the experiments in fig.1
were performed with the tremendous resources of OpenAI. We understand the suggestion to carry out more experiments
for larger model sizes ($10^9$-$10^{11}$), however it comes with a price tag that is unattainable for a small academic research
group (GPT3 is $10^{11}$ and cost 10M\$, T5 has $10^9$ and $10^{10}$ variants that cost 10K-100K\$). Given these training costs,
we see a place for theoretical contributions that provide principles for published experiments and a basis for future
experiments. We are glad for R2's implementation, but since we do not know the experiment details it is hard to
comment on its outcome. Indeed Kaplan et al. employ hyper-parameters tunings (LR, initializations, batch size, etc) as
well as uniquely large datasets in order to demonstrate clean trends in fig.1. However, such large-resource optimization
only provides a cleaner proxy to expressiveness, making these experiments a good fit to support our theory. We leave
analysis of large width impact on optimization for future work, and will add a related paragraph with the suggested
references on the aspect of optimization. Note that a large width allows for model parallelism tricks that are not possible
for large depth, so perhaps the existing training paradigm can be specialized and improved for these cases.

Beyond linear networks being a popular subject of study in the theory literature, we encourage **R3** and **R4** to consider
section 2.3 (recently reinforced by: Katharopoulos et al., arxiv 2006.16236) motivating the practical relevance of the
linear self-attention. Minor questions: *R3: We employ the separation rank as a relevant measure of expressivity
(reflecting input dependencies), and point at supporting empirical evidence. *R2: The separation rank is too large to
measure for L>4, for small real networks it shows compliance with our theory - we will add these experiments in an
appendix. *R2: Theorem 1 is independent on N, as it discusses balanced partitions. *Following structural comments,
we will include a better explanation of the form of $g^L \& C(L)$, formally state the separation rank definition, change the
name "claim 1" into "proposition 1" (full proof is in sec3.2 of the SM), clearly explain R2's interpretation of fig.1 which
is indeed a complementary clarifying view of the figure, crystallize the proof sketch and other more minor corrections.

[Meta-Review · NeurIPS 2020]

All reviewers agree this paper provides rigorous analysis of depth-width tradeoffs of a simplified Transformer model, which explains, to some extent, the observed mild effect of depth on Transformers. While reviewers raised some concerns about lack of more experimental verification of the results all reviewers agree that the paper makes enough interesting theoretical contributions and suggest acceptance. I agree with this and note to authors that it is imperative they make the changes/clarifications as in reviews/response.